# Why Prototypes Collapse: Diagnosing and Preventing Partial Collapse in Prototypical Self-Supervised Learning

**Gabriel Y. Arteaga**[1], **Marius Aasan**[1], **Rwiddhi Chakraborty**[2], **Martine Hjelkrem-Tan**[1], **Thalles Silva**[3], **Michael Kampffmeyer**[2], and **Adín Ramírez Rivera**[1]

[1]University of Oslo    [2]UiT The Arctic University of Norway    [3]University of Campinas

## Abstract

Prototypical self-supervised learning methods consistently suffer from partial prototype collapse, where multiple prototypes converge to nearly identical representations. This undermines their central purpose—providing diverse and informative targets to guide encoders toward rich representations—and has led practitioners to over-parameterize prototype sets or add ad-hoc regularizers, which mitigate symptoms rather than address the root cause. We empirically trace the collapse to the joint optimization of encoders and prototypes, which encourages a type of shortcut learning: early in training prototypes drift toward redundant representations that minimize loss without necessarily enhancing representation diversity. To break the joint optimization, we introduce a fully decoupled training strategy that learns prototypes and encoders under separate objectives. Concretely, we model prototypes as a Gaussian mixture updated with an online EM-style procedure, independent of the encoder's loss. This simple yet principled decoupling eliminates prototype collapse without explicit regularization and yields consistently diverse prototypes, which in several settings translate to improved downstream performance.

## 1 Introduction

Prototypical self-supervised learning (SSL) frameworks (Caron et al., 2021; Siméoni et al., 2025) in the image domain have come to rival the effectiveness of language-supervised alternatives in representation learning (Fan et al., 2025). In these frameworks, prototypes are learnable vectors that act as cluster anchors, guiding learning so that samples organize into semantically coherent regions of the embedding space. However, recent work has demonstrated that many prototypical SSL approaches suffer from a phenomenon known as *partial prototype collapse* (formal definition in Def. 2.1), in which multiple prototypes converge to indistinguishable representations (Govindarajan et al., 2023; 2024). This phenomenon could explain why over-parameterizing the number of prototypes used has been a popular choice to improve performance in recent prototypical frameworks (Oquab et al., 2024; Siméoni et al., 2025; Venkataramanan et al., 2025).

Our experiments show that the practice of *jointly optimizing* the encoder and prototypes contributes to this collapse. The intuition is that joint optimization drives prototypes towards redundant representations early in training by exploiting shortcuts that minimize the loss at the cost of diverse and semantically meaningful representations. This encourages the use of larger prototype sets; an expensive solution that mitigates symptoms without directly addressing the underlying issues (Oquab et al., 2024; Siméoni et al., 2025; Venkataramanan et al., 2025) while impeding a model's ability to learn stronger representations and diminishing robustness to imbalanced data distributions (Govindarajan et al., 2024; Wen et al., 2024).

We find that these effects are particularly pronounced in the *instance-level formulations* of prototypical SSL. In Fig. 1(a), we reaffirm previous findings (Govindarajan et al., 2024), showing how DINO's (Caron et al., 2021) final prototype distribution tends to collapse to one or two modes, which severely limits the diversity of representations. This phenomenon also persists in more recent

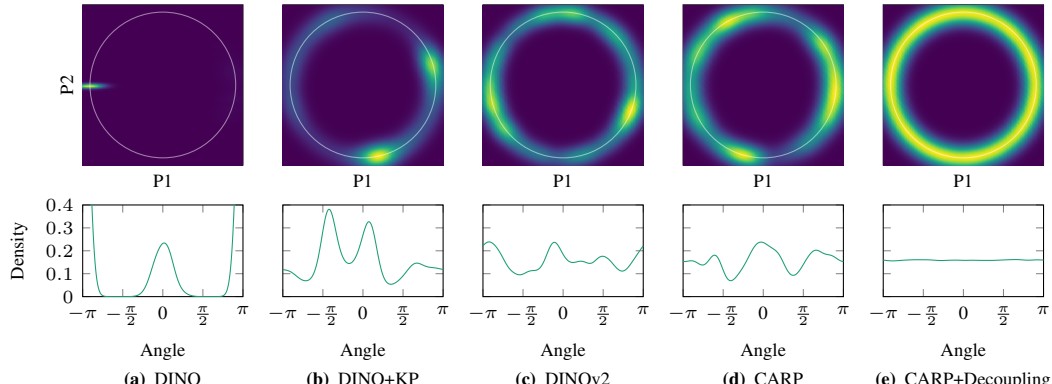

**Figure 1:** Uniformity of prototype representations. Prototypes are projected to $\mathbb{R}^2$ using principal component analysis (PCA). The resulting prototype distributions are visualized using (top) Gaussian kernel density estimation (KDE), and (bottom) the angular distributions are estimated with a von Mises-Fisher KDE with $\kappa = 20$.

methods such as DINOv2 (Oquab et al., 2024), cf. Fig. 1(c). While recent efforts have looked to address this issue, either via explicit regularization (Govindarajan et al., 2024), cf. Fig. 1(b), or implicit mechanisms (Silva & Ramírez Rivera, 2023), cf. Fig. 1(d), these models still exhibit collapse, suggesting that the underlying issue is not yet fully resolved. Motivated by these observations, we introduce a *decoupling strategy* designed to break the joint-optimization procedure. In our approach, the encoder and prototypes are learned under separate objectives—see Fig. 2. Concretely, we model the prototypes as a Gaussian mixture, updated via an expectation maximization (EM) independently of the encoder's loss. *This decoupled training eliminates partial prototype collapse*, cf. Fig. 1(e), and points to joint optimization as the most likely culprit driving partial prototype collapse.

Our main contributions are as follows: (i) we conduct a systematic and quantitative analysis of a broad range of prototypical SSL frameworks, demonstrating that *partial prototype collapse extends well beyond the DINO family of models*; (ii) *we identify the underlying mechanism driving this collapse as the joint optimization of encoders and prototypes under a shared loss*, which encourages redundant prototype representations; and (iii) *we introduce a fully decoupled training framework that isolates prototype estimation from encoder learning, achieving consistently high prototype diversity* throughout training, with downstream representation quality and robustness that improve in several prototypical SSL settings.

## 2 UNDERSTANDING PARTIAL PROTOTYPE COLLAPSE

### 2.1 PRELIMINARIES

**Prototypical Self-Supervised Formulation.** Among recent SSL approaches, prototypical formulations (where network outputs are aligned to a set of learnable, class-agnostic prototypes) have achieved state-of-the-art performance on several vision benchmarks (Siméoni et al., 2025). We begin by describing the general prototypical framework used throughout this paper. Our exposition emphasizes the instance-level objective (Caron et al., 2021; Silva & Ramírez Rivera, 2022; 2023), but the same formulation can readily be extended to incorporate a dense objective (Zhou et al., 2022) or reformulated with an explicit dense prediction objective (Darcet et al., 2025).

Given $J$ stochastic augmentations $\{v^j\}_{j=1}^{J}$ of an image $x$, a student backbone with projection head $f_\theta$ and an Exponential Moving Average (EMA)-updated teacher $f_\phi$ map each view to a latent representation $h_{(\cdot)}^{j} = f_{(\cdot)}(v^j) \in \mathbb{R}^D$. Similarity scores with a learnable prototype set $C \in \mathcal{C} := \mathbb{R}^{D \times K}$ are computed as $z_{(\cdot)}^{j} = h_{(\cdot)}^{j} C \in \mathbb{R}^K$ and converted into prototype-assignment probabilities via a softmax with an optional temperature parameter $\tau > 0$ through

$$p\left(z_{(\cdot)}^{j}\right) = \text{softmax}\left(\frac{z_{(\cdot)}^{j}}{\tau}\right). \tag{1}$$

**Table 1:** We evaluate the number of unique prototypes according to Definition 2.1 with $\epsilon = 0.025$. $^\star$Vanilla iBOT uses the same head for both objectives; for clarity, only one is shown.

| Model | Objective | Init. Protos. | Unique Protos. Dense | Unique Protos. Instance | (% of Init. protos.) |
|---|---|---|---|---|---|
| DINO | Instance | 60000 | – | 908 | (1.5%) |
| CARP | Instance | 65536 | – | 7052 | (10.8%) |
| CAPI | Dense | 16384 | 16383 | – | (99.9%) |
| iBOT | Hybrid | 8192 | 3057$^\star$ | – | (37.3%) |
| iBOT-vMF + KP | Hybrid | 8192 | 7895 | – | (96.4%) |
| DINOv2 | Hybrid | 262144 | 110201 | 2556 | (43.0% ) |

The overall objective of prototypical frameworks is to enforce cross-view consistency between the student and teacher branches using a consistency loss $\mathcal{L}_f$ [1]—see Fig. 2(a).

**Partial Prototype Collapse.** Govindarajan et al. (2024) investigated the DINO-family of methods (Assran et al., 2022; 2023b; Caron et al., 2021; Zhou et al., 2022) and demonstrated that these prototypical SSL approaches are susceptible to *partial prototype collapse*. This phenomenon arises when multiple prototypes converge to nearly identical representations during training. While this behavior does not amount to a complete collapse of all prototypes, it, nevertheless, reduces the diversity of representations and may hinder the effectiveness of downstream tasks. To formalize this behavior, the authors introduced the following definition of partial collapse.

**Definition 2.1** (Partial prototype collapse). Consider the set $C = \{c_k : k = 1, \ldots, K\}$ of $K$ prototype vectors, $c_k$ such that $\|c_k\| = 1$. A *partial prototype collapse* (of degree $M$ and $\epsilon$ distance) is said to have occurred if there exists a set of $M$ disjoint partitions of prototype vectors $V_m \subset C$, $m = 1, \ldots, M$, and $M$ representative prototype vectors $v_m \in V_m$, such that for all $m = 1, \ldots, M$, $1 - v_m^\mathsf{T} c_j < \epsilon$, for all $c_j \in V_m$. The set of $M$ *unique prototypes* is defined as $U = \{v_m\}_{m=1}^M$.

Applying this definition, they further showed that the DINO-family of methods retained only about 2%–40% of their initially initialized prototypes as unique, thereby, revealing a substantial redundancy in the learned representations. To alleviate this issue, they proposed applying KoLeo regularization (Beirlant et al., 1997; Delattre & Fournier, 2017; Sablayrolles et al., 2019) directly to the prototypes. Specifically, they added a KoLeo-Prototype (KP) loss term, $\mathcal{L}_{KP}(C) = -\frac{1}{K} \sum_{k=1}^K \log(d_k)$, where $d_k = \min_{i \neq k} \|c_k - c_i\|$, as an auxiliary term to encourage prototype diversity. While this helped reduce collapse, *it did not fully solve the problem*. Their approach also introduced a new hyperparameter $\lambda_{KP}$, which balances the KP loss with the overall objective, creating a delicate trade-off: insufficient regularization limits diversity, whereas excessive regularization can impair encoder learning.

While the work of Govindarajan et al. (2024) represented an important first step in *identifying* partial prototype collapse, their analysis did not examine its underlying mechanisms and was limited to methods within the DINO family. This opens the door to a broader line of inquiry: *Is partial prototype collapse restricted to the DINO family of methods, and what mechanisms drive this type of collapse?*

## 2.2 DO ALL PROTOTYPICAL SSL FORMULATIONS EXHIBIT PARTIAL PROTOTYPE COLLAPSE?

Despite recent progress, our understanding of prototype collapse remains incomplete. The evidence so far comes exclusively from the DINO family, leaving open whether its partial collapse reflects a universal tendency of prototypical SSL or an artifact of its particular architecture and training regime. To bridge this gap, we next investigate alternative prototypical frameworks, evaluating whether their learned prototypes exhibit similar degrees of redundancy—or if some, by design, avoid collapse altogether.

To this end, we conduct a straightforward analysis of the official weights [2] of several prominent prototypical approaches. We evaluate the diversity of their learned prototypes by setting $\epsilon = 0.025$ [3], following Govindarajan et al.'s (2024) setup and as defined in Definition 2.1. From the results in Table 1, we observe that CARP (Silva & Ramírez Rivera, 2023) achieves substantially higher prototype diversity than DINO. This difference may be explained by its random partitioning strategy,

---

[1] Common choices include the cross-entropy loss (Caron et al., 2021) and the cluster loss (Silva & Ramírez Rivera, 2023).

[2] We consider methods that have made their corresponding *prototype weights* publicly available.

[3] Equivalent to the inspected vectors having at most $12.84°$ between them.

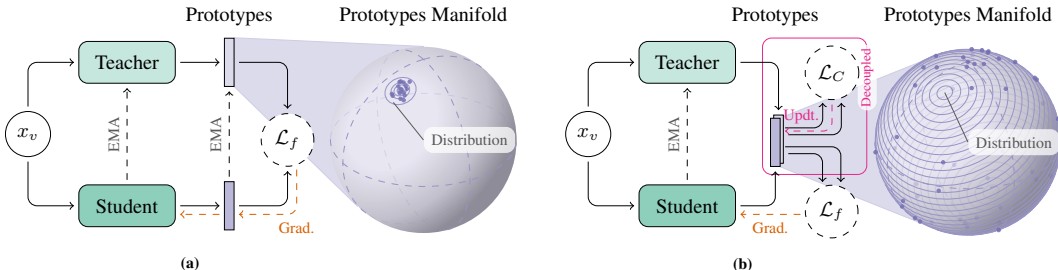

**Figure 2:** (a) Traditional joint-embedding architectures with a prototypical formulation: encoders and prototypes are optimized jointly under the same loss, which can lead to shortcut learning and prototype collapse—prototypes converge to similar representations, reducing the effective representation space. (b) Our proposed solution: decouples the gradient flow to the prototypes and updates them with a separate objective, mitigating shortcut learning and preserving prototype diversity.

which closely resembles the subsampling approach proposed by Wen et al. (2024), which has been shown to enhance robustness to uncurated data. Interestingly, DINOv2's with its separate prototype heads for dense and instance-level objectives, reveal that prototype collapse can be highly objective-specific: its dense head remains relatively diverse (15.9% collapse), but the instance-level head suffers near-total collapse (98%).

To our surprise, we find that CAPI (Darcet et al., 2025) maintains markedly higher prototype diversity than the other methods, exhibiting near-complete uniqueness with only one collapsed prototype out of 16,384. It even manages to surpass iBOT combined with the KoLeo-Proto (KP) regularization (Govindarajan et al., 2024), which explicitly encourages prototype diversity. This begs the question, *what underlying mechanism in CAPI allows the prototypes to remain diverse throughout training?*

A key distinction of CAPI compared to earlier prototypical joint-embedding frameworks is that it *partially decouples* the prototypes from the main loss. In most prototypical formulations—particularly within the DINO family—the prototypes used to compute the teacher's soft assignments are an EMA-updated version of those used for the student, keeping teacher and student prototypes tightly coupled through a joint optimization objective. However, Darcet et al. (2025) argue that, in a masked image modeling (MIM) setting such as iBOT (Zhou et al., 2022), this coupling creates a distributional mismatch: the teacher's prototype assignments are computed on unmasked patches, whereas the student's prototype assignments are computed on masked patches. They argue that this joint optimization under distributional mismatch induces training instabilities.

CAPI addresses the instabilities driven by distributional mismatch, by introducing a *partial decoupling* strategy: the teacher encoder produces latent patch embeddings that are assigned to prototypes learned independently within a separate clustering module, and these assignments then serve as targets for the student. The student branch, in turn, predicts these assignments using its own prototypes, which continue to be updated jointly with the encoder. Although Darcet et al. (2025) proposed this mechanism primarily for stabilizing purposes, *our analysis shows that it also alleviates partial prototype collapse to an extent and, as a result, improves prototype spread.*

## 3 SOLVING PARTIAL PROTOTYPE COLLAPSE THROUGH DECOUPLING

Our analysis of CAPI reveals that decoupling the teacher's prototype updates from the main objective coincides with improved prototype diversity. This finding leads us to hypothesize that *partial prototype collapse in prototypical SSL arises primarily from updating prototypes and encoders under a shared loss*. When prototypes are optimized jointly with the encoder, they may drift toward similar representations that minimize prediction error without improving the underlying features—a form of shortcut learning.[4] Conversely, decoupling the teacher's prototypes from the main loss, as in CAPI (Darcet et al., 2025), should reduce this incentive, leading to more stable and diverse prototype usage throughout training. This observation leads to our formal problem statement:

---

[4]We clarify that by "mitigating shortcut learning" we do not mean shortcuts in relation to spurious correlations and group robustness (Geirhos et al., 2020), but to the early drift of prototypes into redundant representations that artificially reduce the loss without improving the encoder's representations.

**Problem Formulation.** Traditional prototypical SSL methods jointly optimize an encoder $f_\theta$ and a set of prototypes $C = \{c_k\}_{k=1}^K$ by minimizing a consistency loss over augmented views,

$$\min_{\theta, C} \; \mathcal{L}_f\big(f_\theta, C\big). \tag{2}$$

This joint optimization often induces a form of *shortcut learning*, where the prototypes' distribution rapidly collapse into a narrow region of the representation space early in training. Such premature collapse undermines the very purpose of learning the prototypes $C$—to provide diverse and informative targets that guide $f_\theta$ toward representations that transfer well to downstream tasks.

If partial prototype collapse is influenced by the joint optimization of prototypes and encoders, then separating their updates may reduce the likelihood or severity of the collapse. Motivated by this intuition, and unlike CAPI (Darcet et al., 2025), which decouples only the teacher's prototypes from the main loss while keeping the student prototypes jointly optimized, we propose a *fully decoupled* training procedure that isolates prototype estimation from encoder learning, cf. Fig. 2(b), allowing us to directly probe the role of joint optimization in driving collapse.

**Proposed Solution: Full Decoupling.** Instead of jointly solving the original loss (2), over the iterations $t$, we alternate two separate objectives: (i) update the prototypes by solving an independent unsupervised estimation problem on the latent features, i.e.,

$$C^t = \arg\min_{C \in \mathcal{C}} \; \mathcal{L}_C\big(C^{t-1}, h_\phi^t\big), \tag{3}$$

and (ii) update the encoder for fixed prototypes by minimizing the consistency loss, i.e.,

$$\theta^{t+1} = \arg\min_\theta \; \mathcal{L}_f\big(h_\theta^t, C^t\big), \tag{4}$$

where $h^t$ denotes the latent representations at iteration $t$, and $\mathcal{L}_C$ is a loss that estimates prototypes directly from the latent features, independent of the encoder's loss $\mathcal{L}_f$. By fully separating prototype estimation from encoder optimization, our method removes the shortcut incentive and yields stable, diverse prototypes throughout training. A theoretical motivation is provided in Appendix C.1.

### 3.1 DECOUPLING THE OPTIMIZATION

A central question in our framework is how to estimate prototypes once they are decoupled from the encoder. Breaking the joint optimization opens up a rich design space: many unsupervised objectives could, in principle, be used for prototype estimation in Eq. (3). To be effective, however, such objectives must satisfy three key properties: (i) they should be *representative and distinctive*, ensuring that each prototype captures a coherent and separate mode of the data; (ii) they should be *learned over the evolving dataset* rather than isolated mini-batches, to avoid noisy estimates and the resulting training instabilities; and (iii) they should be *computationally efficient*, so as not to impede training time or create memory bottlenecks.

An obvious way to optimize Eq. (3) is to run $K$-Means clustering on the latent features. However, naively applying $K$-Means at every iteration violates two of the key properties outlined above: it bases prototype updates on the current mini-batch rather than the evolving dataset—breaking property (ii)—and it is prohibitively expensive to run online at scale—breaking property (iii). Deep-Cluster (Caron et al., 2018) and PCL (Li et al., 2021) alleviated the cost by clustering only once per epoch on the full dataset, but this left prototypes outdated relative to the changing representation space. SWaV (Caron et al., 2020) addressed the issue of outdated prototypes by introducing an online, gradient-based method for prototype updates; however, this introduced the very joint-optimization dilemma we seek to avoid.

We address the optimization problem of Eq. (3), while satisfying the three outlined properties, by representing the prototypes as the means of an online Gaussian Mixture Model (GMM) (Neal & Hinton, 1998; Sato & Ishii, 2000). In contrast to naive $K$-Means, the online GMM incrementally updates its mixture components as new data arrive, allowing the prototypes to evolve continuously with the representation space. This procedure naturally incorporates information from the entire dataset over time rather than relying on isolated mini-batches, thereby satisfying property (ii), while its incremental updates render it computationally feasible at scale, satisfying property (iii). Moreover, mixture-based clustering has already been successfully applied in several deep learning contexts (Liang et al., 2022; Pu et al., 2023; Zhao et al., 2023), underscoring its potential to satisfy property (i).

While incorporating the GMM into our framework, we developed a variant specifically designed to cope with the high-dimensional feature spaces and large prototype counts that naturally emerge in large-scale representation learning. Without modification, standard online updates at this scale suffer from unbalanced responsibilities and component drift. Drawing on responsibility-weighted forgetting (Celaya & Agostini, 2015) and deterministic annealing (Ueda & Nakano, 1998), we modulate both the strength of parameter updates and the sharpness of assignments. This ensures that components with fewer assignments maintain stable estimates while those with higher usage remain responsive, resulting in a more stable mixture and higher-quality prototypes at scale. We refer the reader to Appendix A for further implementation details.

## 4 EXPERIMENTS

### 4.1 EXPERIMENTAL SETUP

To evaluate our decoupling strategy, we adopt CARP (Silva & Ramírez Rivera, 2023) as our primary point of reference. We select CARP for two main reasons. First, because it is an instance-based approach, and instance-based methods in general exhibit a higher degree of prototypical collapse (cf. Table 1), CARP provides a more challenging setting for evaluating our decoupling method, while also isolating the effect of MIM objectives on prototype diversity, which we leave for future work. Second, extensive ablation experiments have demonstrated that CARP maintains robust and stable performance across a wide range of hyperparameters (Silva & Ramírez Rivera, 2023), making it a suitable framework for evaluating our decoupling strategy without extensive hyperparameter tuning. In addition, we evaluate DINO (Caron et al., 2021) with our decoupling method, incorporating DINO enables a broader assessment of the generality of our method and its impact on prototypical diversity.

Except for the experiment reported in Section 4.2, where we rely on the officially released weights, we train all models ourselves.[5] This enables us to analyze the training dynamics of the different methods in greater depth (an analysis that would not be possible without access to intermediate checkpoints) and to provide a more detailed assessment of how prototype diversity benefits the handling of long-tailed distributions.

### 4.2 UNIQUENESS OF PROTOTYPES UNDER DIFFERENT THRESHOLDS

To test our hypothesis *that joint optimization is responsible for partial prototype collapse* (*and that decoupling prevents it*) we evaluate prototype diversity by counting the number of unique prototypes under different $\epsilon$ configurations. Setting $\epsilon = 0$ imposes no restriction and therefore shows the total number of initialized prototypes. Govindarajan et al. (2024) evaluated unique prototypes using $\epsilon = 0.025$, which we also showcase. To examine partial prototype collapse more closely and to assess the effectiveness of our decoupling approach, we additionally evaluate using a stricter threshold of $\epsilon = 0.5$, i.e., 20 times stricter than that of Govindarajan et al.'s (2024) setup, this is equivalent to the prototypes being unique only if they are separated by at least $60°$.

By examining Fig. 3(a), several notable insights emerge. First, a substantial collapse is observed in DINO and DINOv2, highlighting the extent of collapse within the DINO family of methods when no preventive measures are applied. Second, CAPI preserves prototype diversity at lower thresholds of $\epsilon$, exhibiting only minor collapse. As the constraint increases, CAPI still retains approximately $38\%$ of its initialized prototypes. *We attribute CAPI's robustness to prototypical collapse to its partial decoupling mechanism*, in which the teacher-branch prototypes are updated using a loss function separate from that of the student-branch encoder.

Lastly, we *observe no evidence of partial prototype collapse* across all tested values of $\epsilon$ when the prototype objective is *fully decoupled* from the overall loss. This stands in clear contrast to CARP, which exhibits collapse in $90\%$ of its prototypes already at $\epsilon = 0.025$ and CAPI with its *partial decoupling*. Crucially, our approach not only prevents partial prototype collapse but also *improves the encoder's learned representations*, as demonstrated in Fig. 3(b). This distinction is central to our contribution: it supports our claim that joint optimization introduces a form of shortcut learning (minimizing the loss without genuinely enriching the encoder's representations) whereas decoupling avoids this pitfall and yields more informative features.

---

[5]Details of how the baselines were trained are provided in Appendix B.

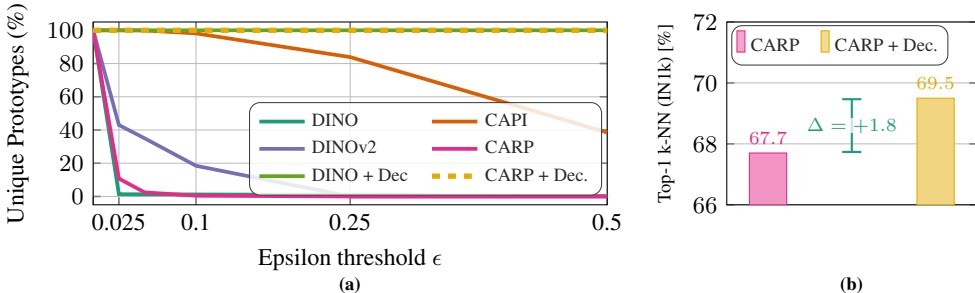

**Figure 3:** (a) Unique prototypes versus $\epsilon$ (Definition 2.1). At $\epsilon = 0$ all initialized prototypes are counted as unique, whereas increasing $\epsilon$ enforces stricter criteria (e.g., at $\epsilon = 0.5$ only prototypes separated by at least $60°$ are considered unique). (b) k-NN performance on IN1k for CARP showcasing an improvement with decoupling.

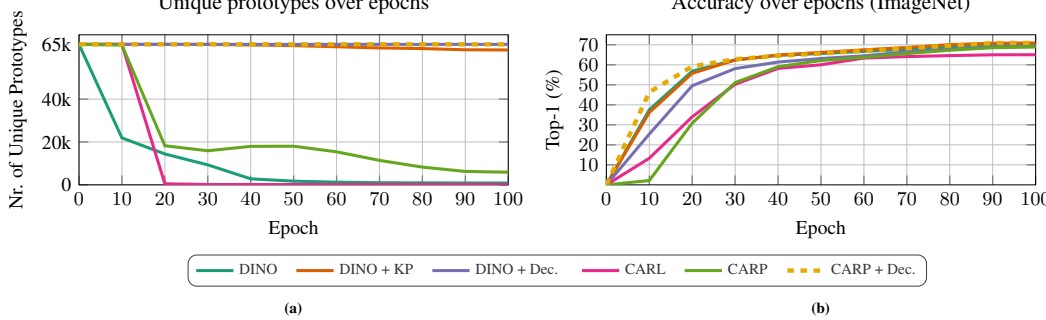

**Figure 4:** (a) Number of unique prototypes over epochs. Prototypical methods without explicit diversity mechanisms (e.g., DINO and CARL) exhibit substantial prototype collapse early in training. (b) Linear evaluation accuracy on ImageNet-1k. Higher prototype diversity generally correlates with stronger final performance, though some methods (e.g., DINO + KP) imposes a cost of slower early convergence.

## 4.3 TRAINING DYNAMICS

Although previous studies have examined partial prototype collapse, they have focused almost exclusively on its manifestation at the end of training (Govindarajan et al., 2023; 2024), leaving its interaction with the training dynamics largely unexplored. To address this and gain deeper insight into the relationship between prototype collapse and training progression, in Fig. 4, we track the number of unique prototypes in use and the linear evaluation accuracy on ImageNet-1k (Deng et al., 2009) over the first 100 epochs for multiple methods. Specifically, we include CARL (Silva & Ramírez Rivera, 2022) and CARP (Silva & Ramírez Rivera, 2023) (CARP being an extension of the CARL framework that introduces random partitioning) alongside DINO (Caron et al., 2021), DINO + KoLeoProto (KP) (Govindarajan et al., 2024), and our proposed decoupled approach.

A first observation is that partial prototype collapse emerges very early in training: after only 10 epochs, two-thirds of the prototypes have already collapsed. Introducing the KP regularizer (Govindarajan et al., 2024) proves effective at maintaining prototype diversity, underscoring its role as a strong collapse-prevention mechanism. While DINO+KP exhibits a modest drop in accuracy during the early stages of training, this effect diminishes over time, and the final model slightly outperforms vanilla DINO. This suggests that preserving higher prototype diversity throughout training ultimately leads to stronger representations in the converged model.

Another noteworthy observation is that CARP, is far less prone to prototype collapse than the baseline CARL. By the end of training, CARL retains only about $0.2\%$ of its initialized prototypes as unique, whereas CARP preserves $9\%$. This difference is also reflected in final linear evaluation performance: CARP outperforms CARL late in training but is impaired early in training. While these gains cannot be attributed solely to prototype uniqueness (training stability likely plays a significant role) they nevertheless amount to a substantial margin of roughly 4 percentage points in linear accuracy by the end of training.

Finally, we find that CARP + Decoupling maintains full prototype utilization while outperforming all other methods in the early and late phases of training, indicating that *decoupling does not hinder its early training dynamics*. By the end of training, CARP + Decoupling improves upon CARP, demonstrating that enhanced prototype diversity yields stronger representations.

## 4.4 Effect of prototype diversity across head, medium and tail classes

Prototypical SSL methods generally perform well on carefully curated datasets; however, their effectiveness deteriorates markedly when trained on uncurated data (Oquab et al., 2024; Siméoni et al., 2025), which commonly exhibit class imbalances and long-tailed distributions characteristic of real-world scenarios (Mahajan et al., 2018; Newman, 2005; Van Horn et al., 2018). We argue, therefore, that a desirable property of any pre-training framework is *robustness to long-tailed data distributions*. Such robustness would enable practitioners to improve learned representations simply by scaling the amount of training data. Nevertheless, Fan et al. (2025) report that scaling up the amount of training data for a 7B-parameter version of DINOv2 yielded limited benefits: as the size of the training data increased, performance across a range of benchmarks decreased. These findings on *data scaling underscore a fundamental limitation of current prototypical SSL approaches*.

While Govindarajan et al. (2024) and Wen et al. (2024) empirically identified a link between prototype diversity and improved performance on long-tailed distributions, an important question remains: *is the observed improvement primarily due to a better ability to model tail classes, or is it simply a byproduct of improved modeling of head-class data?* To address this question,

**Table 2:** Performance comparison on the iNaturalist 2018 dataset. Results are reported as Top-1 accuracy (%) for different class splits.

| Methods | Head ($> 100$) | Medium ($> 20 \ \& \leq 100$) | Tail ($\leq 20$) | All |
|---|---|---|---|---|
| DINO | 55.2 | 46.2 | 41.9 | 45.3 |
| DINO + KP | 59.8 ↑ 4.6 | 50.4 ↑ 4.2 | 45.0 ↑ 3.1 | 49.0 ↑ 3.7 |
| DINO + Decoupling | 44.9 ↓ 10.3 | 37.3 ↓ 8.9 | 33.1 ↓ 8.8 | 36.2 ↓ 9.1 |
| CARP | 56.0 | 46.9 | 42.5 | 45.9 |
| CARP + KP | 55.5 ↓ 0.5 | 46.4 ↓ 0.5 | 41.8 ↓ 0.7 | 45.3 ↓ 0.6 |
| CARP + Decoupling | 59.1 ↑ 3.1 | 49.3 ↑ 2.4 | 45.9 ↑ 3.4 | 48.9 ↑ 3.0 |

we conduct a granular experiment in which we train DINO and CARP with KoLeo-Proto (KP) regularization and our proposed full decoupling mechanism on the iNaturalist-18 dataset (Van Horn et al., 2018). This dataset contains approximately 430K images spanning 8,142 classes, with a naturally long-tailed class distribution, making it an ideal benchmark for assessing the robustness of different methods under realistic data conditions. Following Liu et al.'s (2019) definition of class subcategories, we define head classes as those with more than 100 training instances, tail classes as those with fewer than 20, and medium classes as those in between. This setup provides a detailed view of how each method performs across the head, medium, and tail classes, and offers insights into the benefits stemming from increased prototype diversity.

Table 2 presents the performance of all methods across the head, medium, and tail splits of iNaturalist 2018. Consistent with prior observations (Govindarajan et al., 2024), increasing prototype diversity often improves robustness under long-tailed class distributions. In addition, *this experiment shows that these gains are not confined to head classes: improvements appear across all class-frequency regimes, including the tail classes.* For DINO, KP regularization yields a 3.7-percentage-point gain in overall accuracy. In contrast, applying our decoupling strategy directly to DINO leads to a substantial performance drop across all class-frequency regimes. We attribute this decline to the high sensitivity of DINO to excessive prototype diversity; in ImageNet-1k, extensive hyperparameter tuning was required to stabilize DINO under decoupling, and

**Table 3:** ImageNet evaluation for instance-based prototypical SSL methods.

| Method | Backbone | Epochs | k-NN (%) | Linear (%) |
|---|---|---|---|---|
| DINO | RN-50 | 400 | 67.5 | 75.3 |
| SWAV | RN-50 | 400 | 65.0[†] | 74.6 |
| DeepCluster-v2 | RN-50 | 800 | 66.6[†] | 75.2 |
| CARP | RN-50 | 400 | 67.7 | 75.3 |
| CARP + Decoupling | RN-50 | 400 | 69.1 | 75.3 |
| DINO | ViT-S/16 | 300 | 72.8 | 76.2 |
| CARP | ViT-S/16 | 300 | 73.6 | 76.3 |
| CARP + KP | ViT-S/16 | 300 | 73.7 | 76.1 |
| CARP + Decoupling | ViT-S/16 | 300 | 74.1 | 76.2 |
| MSN | ViT-S/16 | 600 | –[‡] | 76.9 |
| DINO | ViT-S/16 | 800 | 74.5 | 77.0 |
| DINO-vMF | ViT-S/16 | 800 | 74.7 | 77.0 |
| CARP + Decoupling | ViT-S/16 | 800 | 75.3 | 76.4 |
| DINO | ViT-B/16 | 400 | 76.1 | 78.2 |
| DINO-vMF | ViT-B/16 | 400 | 77.4 | 78.8 |
| CARP + Decoupling | ViT-B/16 | 400 | 76.7 | 78.1 |

[†]Results computed by us using the officially released pre-trained models.
[‡]Official pre-trained weights are not publicly available

these settings do not transfer effectively to the heavily imbalanced iNaturalist domain. A detailed analysis of DINOs hyperparameter sensitivity is provided in Appendix C.3.

For CARP, the trends differ: adding KP results in a slight decrease in accuracy, which we suspect is linked to the choice of KP regularization strength. Our experiments adopt the value recommended by Govindarajan et al. (2024), selected based on ablations conducted for DINO, an architecture exhibiting substantially stronger prototypical collapse than CARP, as demonstrated in Figure 3(a). When applied to CARP under long-tailed conditions, using the same strength appears to hinder learning. In contrast, integrating our decoupling strategy with CARP yields consistent improvements across all frequency regimes, amounting to a 3.0-percentage-point gain overall.

In summary, and in line with Govindarajan et al. (2024), increasing prototype diversity in prototypical SSL frameworks may substantially improve robustness under long-tailed data distributions. *Our*

**Table 4:** Transfer learning results. We follow Ericsson et al. (2021) transfer learning protocol. Top performing in **bold**, top-2 underlined.

| Method | Epochs | Aircr | C101 | Cars | Flower | Food | Pets | SUN397 | VOC2007 | Avg |
|---|---|---|---|---|---|---|---|---|---|---|
| **ResNet-50** | | | | | | | | | | |
| DINO | 400 | 59.95 | 90.91 | 65.92 | 95.63 | 78.40 | 89.04 | 66.08 | 84.32 | 78.78 |
| CARP | 400 | **61.03** | 91.66 | 64.21 | 95.82 | **78.88** | 90.25 | **66.10** | 84.51 | 79.06 |
| CARP + Decoupling | 400 | 58.50 | **92.07** | **67.28** | **95.91** | 78.74 | **91.11** | 65.24 | **84.55** | **79.18** |
| **ViT-Small/16** | | | | | | | | | | |
| CARP | 300 | 59.16 | **93.61** | **64.48** | 96.40 | 77.51 | 93.38 | 65.94 | 85.18 | 79.46 |
| CARP + KP | 300 | 60.48 | 93.54 | 63.55 | **96.44** | **78.61** | 93.40 | **65.98** | 85.34 | **79.67** |
| CARP + Decoupling | 300 | **61.78** | 92.91 | 63.90 | 96.20 | 78.49 | **94.13** | 64.53 | **85.35** | 79.66 |

*granular analysis confirms that these gains extend consistently to tail classes rather than being driven solely by improved modeling of head classes.* However, for DINO, an excessively large degree of prototype diversity appears to render optimization unstable and requires extensive hyperparameter adjustment. Likewise, for KP, our experiments indicate that careful tuning of the regularization strength across frameworks is necessary when operating in long-tailed settings.

## 4.5 Linear Classification on ImageNet-1k

We obtain strong k-NN performance with our proposed decoupling approach built on top of CARP, demonstrating the benefit of increased prototype spread with a 1.8-percentage-point improvement in k-NN performance (Table 3). As is well known (Caron et al., 2021), linear evaluation results are highly sensitive to hyperparameter choices, particularly for methods with diverse prototypes (Darcet et al., 2025; Oquab et al., 2024). Consequently, rather than performing extensive hyperparameter tuning for linear classification, we follow a light grid-search protocol[6] and place primary emphasis on the k-NN metric, which provides a more robust, fine-tuning-free assessment of representation quality.

## 4.6 Transfer Learning

The transfer learning results in Table 4 show that adding decoupling to CARP does not lead to a substantial increase in transfer learning performance. Across ResNet-50 and ViT-Small/16 backbones, the decoupled variants achieve comparable accuracy to the standard CARP, with small variations across datasets. This is consistent with prior observations by Govindarajan et al. (2024), who report that increasing prototype diversity does not necessarily lead to improved transfer performance.

## 4.7 Memory Usage and Time Consumption

To assess the memory implications of the proposed decoupling strategy, we measure peak GPU memory reserved per iteration when training a ViT-S/16 model on ImageNet-1k with 10 local crops, comparing the joint-optimized CARP baseline with our decoupled variant while varying the global batch size. All experiments are conducted on AMD Instinct MI250X GPUs. Table 5 reports the per-GPU memory reserved for each approach. At smaller batch sizes (e.g., 128), the reduction in memory footprint is minimal, as activation and optimizer-state requirements dominate overall usage in this regime. However, as the batch size increases, the decoupled formulation yields progressively larger savings. This behavior follows from the mechanics of the online GMM update: because prototype parameters are updated immediately after the forward pass and independently of the encoders loss, the method avoids storing the computation graph and intermediate tensors associated with prototype optimization. In contrast, joint-optimization approaches must retain these tensors until backpropagation, contributing to the observed increase in memory usage with larger batch sizes. Because patch-level objectives introduce substantially higher memory requirements, we expect the memory savings from decoupling to be even greater in methods such as iBOT or DINOv2, which we leave as an area for future investigation.

**Table 5:** Peak memory reserved per GPU for CARP and CARP + Dec. across increasing global batch sizes.

| Batch Size | CARP | CARP + Dec. |
|---|---|---|
| 128 | 5.4G | 5.3G ↓0.1 |
| 256 | 9.3G | 9.1G ↓0.2 |
| 512 | 17.3G | 16.8G ↓0.5 |
| 1024 | 32.3G | 31.5G ↓0.8 |
| 2048 | 62.4G | 60.9G ↓1.5 |

In terms of training time, Table 6 shows that the proposed decoupling method adds only a minimal increase, just 0.2 hours, indicating that the memory savings are achieved without any significant impact on runtime.

---

[6]See Appendix C.2.1 for an extended discussion of the linear evaluation protocol and additional results.

# 5 RELATED WORK

**Self-Supervised Learning** leverages unlabeled data by enabling the model to generate its own supervisory signals. Early approaches introduced pretext tasks such as image inpainting (Pathak et al., 2016) and solving jigsaw

**Table 6:** Training time comparison showing that introducing the online Gaussian Mixture Model for decoupling incurs only a negligible increase of 0.2h.

| Method | Epochs | Batch Size | Total Crops | Time (h) |
|---|---|---|---|---|
| CARP | 100 | 1024 | 12 | 37.9 |
| CARP + Decoupling | 100 | 1024 | 12 | 38.1 |

puzzles (Noroozi & Favaro, 2016). Since then, a dominant paradigm has emerged that employs joint-embedding architectures within a contrastive framework (Chen et al., 2020; Chen & He, 2021; Chen et al., 2021; Grill et al., 2020), which was later extended by incorporating a predictor network (JEPA) (Assran et al., 2023a). Another line of work employs reconstruction-based objectives instead (Bao et al., 2022; He et al., 2022).

**Prototypical Self-Supervised Learning** extends joint-embedding methods by introducing a discrete set of learnable prototypes that serve as targets for representation assignment. Such formulations have achieved state-of-the-art performance among vision-only SSL approaches (Siméoni et al., 2025; Venkataramanan et al., 2025), rivaling the effectiveness of language-supervised alternatives (Fan et al., 2025; Radford et al., 2021). Early methods performed cluster assignments once per epoch (Caron et al., 2018; Li et al., 2021), which could result in outdated prototypes. To mitigate this issue, online methods were introduced (Asano et al., 2020; Caron et al., 2020), enabling the joint optimization of the encoder and prototypes, a strategy that has since become the status quo for most prototypical formulations (Assran et al., 2022; 2023b; Caron et al., 2021; Oquab et al., 2024; Ruan et al., 2023; Silva & Ramírez Rivera, 2022; Siméoni et al., 2025; Zhou et al., 2022). However, this joint optimization also introduced the phenomenon of partial prototype collapse which Govindarajan et al. (2023) was first to identify, and has since been addressed to varying degrees of success both explicitly (Govindarajan et al., 2024; Wen et al., 2024) and implicitly (Darcet et al., 2025; Silva & Ramírez Rivera, 2023). Recent work departs from *learning* prototypes explicitly, instead using fixed high-dimensional codes sampled from a Rademacher distribution (Sansone et al., 2025) or non-parametric targets built from a queue of past encoder representations that act as effective prototypes (Gidaris et al., 2021; 2024; Silva et al., 2024; 2025).

**Limitations:** Although our proposed decoupling approach alleviates prototype collapse, it introduces several limitations. The current implementation lacks a mechanism to ensure that prototype updates track the encoders rate of change, as updates rely solely on the forgetting factor $\eta$, which cannot adapt to shifts in the online estimation of the mixture. The use of an online GMM also adds hyperparameters, and it assumes that incoming representations follow a Gaussian distribution. While our experiments show that the method eliminates prototype collapse across all investigated thresholds, the complete removal of collapse is not universally beneficial across frameworks (cf. Appendix A.3).

# 6 CONCLUSION

This work provides, to the best of our knowledge, the first systematic investigation into the root causes of partial prototype collapse in prototypical self-supervised learning. Through an extensive empirical analysis across a diverse set of frameworks, we show that prototype collapse is not confined to the DINO family but is a widespread phenomenon, drastically reducing the effective diversity of prototypes and limiting downstream transfer performance. We *identify joint optimization of encoders and prototypes under a shared loss* as the key mechanism driving this collapse-encouraging a type of shortcut learning and redundant prototype representations early in training.

Building on this diagnosis, we *introduce a fully decoupled training framework* that isolates prototype estimation from encoder learning. By representing prototypes as a continuously updated Gaussian mixture and estimating them via an online EM-style procedure, our approach removes the shortcut incentive inherent to joint optimization. This design yields *stable and highly diverse prototypes throughout training* without ad-hoc regularization or hyperparameter trade-offs.

Across extensive experiments, including imbalanced datasets, our decoupling strategy consistently enhances prototype spread, while its impact on downstream representations and robustness to challenging data distributions varies across architectures. Together, these findings highlight the benefits of breaking the joint optimization paradigm, demonstrating that decoupled prototype learning is a simple, principled, and scalable strategy for preventing collapse and improving representation quality in several prototypical SSL settings.

## ACKNOWLEDGMENTS

This work was funded, in part, by RCN (the Research Council of Norway) through Visual Intelligence, Centre for Research-based Innovation (grant no. 309439), and FRIPRO (grants no. 315029 and 359216). The computations were performed, in part, on resources provided by Sigma2—the National Infrastructure for High-Performance Computing and Data Storage in Norway (Project NN8104K). We thank Shashanka Venkataramanan and Yuki M. Asano (Franca team) for providing their reproduced DINOv2 prototype weights, which broadened the scope of our study.

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

## A  ONLINE GAUSSIAN MIXTURE MODEL

To decouple the joint optimization of the encoder and prototypes, we choose to represent prototypes explicitly as components of a Gaussian Mixture Model (GMM). Under this probabilistic formulation, prototypes capture the underlying structure of the data distribution independently from the encoder's training objective. By leveraging an Expectation-Maximization (EM)-based approach (Dempster et al., 1977), prototypes maximize the data likelihood of the teacher's latent embeddings, thereby, effectively disentangling their update mechanism from the encoder's loss-driven gradient updates. This separation ensures that prototype optimization is solely governed by the statistical characteristics of the latent representations ($\mathcal{L}_C$), promoting robust learning and preventing partial collapse.

While the classical EM algorithm provides a principled framework for prototype learning, it assumes access to a static and complete dataset. This assumption is incompatible with most SSL frameworks, where data becomes available incrementally as the encoder continuously updates its representations during training. To address this limitation, we propose to adopt an online variant of the EM algorithm (Neal & Hinton, 1998; Sato & Ishii, 2000), which updates the mixture parameters incrementally using mini-batches of data.

At each iteration $t$, we treat the incoming batch as a sample from the evolving feature distribution and perform a two-step update: computing responsibilities for each latent vector, followed by an update of the GMM parameters using sufficient statistics.

The GMM is parameterized by $\Psi^{(t-1)} = \{\pi^{(t-1)}, \mu^{(t-1)}, \Sigma^{(t-1)}\}$, where $\pi$, $\mu$, and $\Sigma$ denote the mixture weights, means (prototypes), and diagonal covariances, respectively. For each input $i$, the responsibility $\gamma_{ik}^{(t)}$ is defined as:

$$\gamma_{ik}^{(t)} = \frac{\pi_k^{(t-1)} P\left(h_i^{(t)} \mid \mu_k^{(t-1)}, \Sigma_k^{(t-1)}\right)^{\beta}}{\sum_{k'=1}^{K} \pi_{k'}^{(t-1)} P\left(h_i^{(t)} \mid \mu_{k'}^{(t-1)}, \Sigma_{k'}^{(t-1)}\right)^{\beta}}. \tag{A.1}$$

Here, $\gamma_{ik}^{(t)}$ denotes the soft assignment of latent vector $h_i^{(t)} = f_\phi^{(t)}(v_i)$ from the teacher network to the $k$-th Gaussian component. Intuitively, it reflects how well the $k$-th component's distribution explains $h_i^{(t)}$. To further enhance prototype utilization, particularly when employing a large number of components $K$, we introduce an annealing mechanism controlled by a smoothing factor $\beta \in [0, 1]$ (Ueda & Nakano, 1998), which modulates the sharpness of the responsibilities during training.

Following the assignment of responsibilities, the expectation step utilizes these values to compute the intermediate sufficient statistics for the current timestep necessary to update the mixture's sufficient statistics. Specifically, at each time step, the intermediate sufficient statistics for a batch $B$ and views $J$ are calculated as:

$$\tilde{S}_k^{(t)} = \left\{\tilde{S}_{k,\pi}^{(t)},\ \tilde{S}_{k,\mu}^{(t)}, \tilde{S}_{k,\Sigma}^{(t)}\right\} = \left\{\sum_{i=1}^{JB} \gamma_{ik}^{(t)}, \sum_{i=1}^{JB} \gamma_{ik}^{(t)} h_i^{(t)}, \sum_{i=1}^{JB} \gamma_{ik}^{(t)} h_i^{(t)} h_i^{(t)^T}\right\}. \tag{A.2}$$

These intermediate statistics are then used to update the sufficient statistics of the mixture using a weighted moving average:

$$S_k^{(t)} = \begin{cases} \eta^{\hat{\gamma}_k} S_k^{(t-1)} + \left(1 - \eta^{\hat{\gamma}_k}\right) \tilde{S}_k^{(t)}, & \text{if } t > 1, \\ \tilde{S}_k^{(1)}, & \text{otherwise.} \end{cases} \tag{A.3}$$

$$\text{where} \quad \tilde{S}_k^{(1)} = \left\{ \frac{JB}{K}, \ \mu_k^{(1)} \tilde{S}_{k,\pi}^{(1)}, \ \Sigma_k^{(1)} \tilde{S}_{k,\pi}^{(1)} + \frac{\tilde{S}_{k,\mu}^{(1)} \tilde{S}_{k,\mu}^{(1)\top}}{\tilde{S}_{k,\pi}^{(1)}} \right\}. \tag{A.4}$$

Here, $\eta \in [0, 1]$ is a forgetting factor, controlling the influence between new and past information. When using a large number of prototypes $K$ the mixture's components will receive sparse updates, thus, the forgetting factor $\eta$ will tend to push the sufficient statistics towards zero. To circumvent this issue, we introduce a responsibility based forgetting mechanism (Celaya & Agostini, 2015) through the incorporation of the expectation of the responsibilities over the batch dimension $\hat{\gamma}_k = \frac{1}{JB} \sum_i^{JB} \gamma_{ik}$. By raising the forgetting factor to the power of $\hat{\gamma}_k$, components with higher expected responsibility receive more aggressive updates, whereas those with lower responsibilities retain a larger fraction of their previous statistics, thereby, preventing the sufficient statistics to be pushed towards zero.

Given the updated sufficient statistics, we proceed to the maximization step of the EM algorithm, wherein the GMM parameters $\Psi$ are updated. This step updates the mixture weights, means, and covariances by computing their maximum likelihood estimates based on the current sufficient statistics:

$$\pi_k^{(t)} = \frac{S_{k,\pi}^{(t)}}{\sum_{k'=1}^{K} S_{k',\pi}^{(t)}}, \quad \mu_k^{(t)} = \frac{S_{k,\mu}^{(t)}}{S_{k,\pi}^{(t)}}, \quad \Sigma_k^{(t)} = \frac{S_{k,\Sigma}^{(t)}}{S_{k,\pi}^{(t)}} - \frac{S_{k,\mu}^{(t)} S_{k,\mu}^{(t)T}}{S_{k,\pi}^{(t)} S_{k,\pi}^{(t)}}. \tag{A.5}$$

In the context of our prototypical SSL framework, the updated means $\mu_k$ serve as prototypes for the latent assignments, i.e., $\mu_k = c_k$ for $k = 1, \ldots, K$. It is important to note that this maximization step is performed prior to the encoder's optimizer step, ensuring that the prototypes reflect the most recent assignment statistics before any encoder parameters are updated.

Throughout training, we use a variation of split-and-merge (Ueda & Nakano, 2000) for regularization. But instead of merging low-weight clusters, we instead we apply a method we call *split-resurrect*. Whenever some component $k$ exceeds a weight threshold, we identify the lightest component $j$, reinitialize its mean (scale-aware random initialization) and reset its variance, and then split the mass of $k$ evenly, $\pi_k \leftarrow \pi_j \leftarrow \frac{1}{2}\pi_k^{\text{old}}$. For high cluster weights, we regularize the scale of the dominant mean by

$$\hat{\mu}_k^{(t)} = \mu_k^{(t)} / \sqrt{\|\mu_k^{(t)}\|_2} \tag{A.6}$$

which avoids dominant high-norm clusters in the GMM.

## A.1 ABLATION: IMPORTANCE OF THE DIFFERENT COMPONENTS

We assess the impact of the individual components in our proposed decoupling strategy — the online Gaussian mixture. Specifically, we train CARP with a ViT backbone using 10 local crops on ImageNet-1k to evaluate each component. The linear accuracy is reported using PyTorchs L-BFGS solver (Liu & Nocedal, 1989; Paszke et al., 2019).

As shown in Table A.1, the responsibility-forgetting mechanism introduced by Celaya & Agostini (2015) is essential given the sparse updates that occur when using a large number of prototypes. Without this mechanism, the forgetting factor $\eta$ drives the sufficient statistics toward zero, ultimately leading to collapse. While the remaining modifications are less detrimental, they nonetheless yield measurable gains in the encoder's representational quality to varying degrees.

## A.2 ABLATION: SENSITIVITY ANALYSIS OF THE FORGETTING-FACTOR SCHEDULER

The forgetting factor $\eta$ used in the online GMM update is defined through a scheduler of the form

$$\eta = 0.99 - \frac{1}{at - b}, \tag{A.7}$$

**Table A.1:** Ablation study of model components.

| Responsibility Based Forgetting (Celaya & Agostini, 2015) | Annealing (Ueda & Nakano, 1998) | Resurrect | Rescaling | Lin. |
|:---:|:---:|:---:|:---:|:---:|
| ✗ | ✓ | ✓ | ✓ | 0.1 |
| ✓ | ✗ | ✓ | ✗ | 71.65 |
| ✓ | ✗ | ✓ | ✓ | 71.97 |
| ✓ | ✗ | ✗ | ✗ | 71.99 |
| ✓ | ✓ | ✗ | ✗ | 72.12 |
| ✓ | ✓ | ✓ | ✓ | **72.25** |

**Table A.2:** Ablation of forgetting-factor scheduler parameters on ImageNet-1k with CARP+Dec..

| Method | Arch | Epoch | $a$ | $c$ | k-NN (%) |
|:---|:---|:---|:---|:---|:---|
| CARP + Dec. | ViT-S/16 | 100 | 0.001 | 0.75 | 66.82 |
| CARP + Dec. | ViT-S/16 | 100 | 0.001 | 0.95 | 67.41 |
| CARP + Dec. | ViT-S/16 | 100 | 0.0005 | 0.89 | 69.26 |
| CARP + Dec. | ViT-S/16 | 100 | 0.002 | 0.89 | 68.99 |
| CARP + Dec. | ViT-S/16 | 100 | 0.001 | 0.89 | **69.52** |

where $t$ denotes the training timestep, $a$ controls the rate at which ($\eta$) approaches its upper bound of 0.99, and $b = \frac{1}{0.99-c}$. The parameter $c$ determines the initial value of the scheduler, while $a$ governs its progression over the course of training.

In the main experiments, we set $a = 0.001$ and $c = 0.89$, which yields an initial forgetting factor of $\eta = 0.89$ at the first training step. To assess the sensitivity of this choice, we evaluated alternative schedules with different values of $a$ and $c$ on ImageNet-1k using a ViT-S trained for 100 epochs with 6 local crops, varying each parameter independently while keeping all other hyperparameters fixed.

Overall, the results indicate that performance remains stable across a reasonable range of scheduler configurations. Adjusting the intercept $c$ produces the largest effect: decreasing $c$ to 0.75 leads to a notable drop in accuracy, as the resulting schedule causes the prototypes to update too aggressively early in training. Increasing $c$ to 0.95 slows the early updates and yields a milder degradation, but still performs worse than the baseline. The intermediate value $c = 0.89$ provides a balanced update rate that appears most effective. In contrast, modifying the convergence rate through $a$ has a more negligible influence on performance, suggesting that the schedule is less sensitive to this parameter.

## A.3 LIMITATIONS

Although our proposed decoupling approach alleviates prototypical collapse, it comes with certain limitations. First, the current implementation lacks a mechanism to ensure that prototype updates remain aligned with the encoders rate of change. Prototype updates are governed solely by the forgetting factor $\eta$, and although a linear schedule performs reasonably well, it does not allow the method to adjust adaptively to the online estimation of the Gaussian mixture. Because $\eta$ plays a role analogous to a learning rate in gradient-based optimization, an interesting direction for future work would be to incorporate momentum to better regulate prototype updates.

Furthermore, incorporating an online GMM to enable our decoupling strategy introduces additional hyperparameters, such as the smoothing factor $\beta$ and the forgetting factor $\eta$, which are required to manage the sparse update regime associated with the large prototype sets used in prototypical SSL frameworks. The online mixture model also assumes that incoming representations follow a Gaussian distribution with meaningful variation in magnitude. This assumption aligns with CARP (Silva & Ramírez Rivera, 2023), which does not normalize its projection head outputs, but it does not strictly hold for DINO (Caron et al., 2021), where the outputs are normalized. Although our internal experiments did not reveal noticeable differences when replacing the Gaussian with a von Mises-Fisher distribution, the current formulation of our decoupling method is built around a Gaussian assumption.

Lastly, our experiments in Section 4.2 and Section 4.3 show that the proposed decoupling method effectively eliminates prototype collapse across all our investigated thresholds $\epsilon$ following Definition 2.1. However, the absence of prototype collapse is not universally beneficial for all frameworks, as

**Table B.1:** Training hyperparameters for ViT-Small configurations for CARP & CARP + Decoupling and ViT-Base configurations forCARP + Decoupling.

**(a)** ViT-Small Configuration

| argument | value |
|---|---|
| architecture | vit_small |
| n_prototypes | 65536 |
| patch size | 16 |
| momentum_teacher | 0.992 |
| drop_path_rate | 0.1 |
| bottleneck_dim | 256 |
| global_crops_scale | [0.32, 1.0] |
| local_crops_scale | [0.05, 0.32] |
| optimizer | AdamW |
| learning rate (lr) | $5e-4$ |
| weight_decay | 0.04 |
| weight_decay_end | 0.4 |
| freeze_prototypes | false |
| min_lr | $1e-6$ |
| clip_grad | 1.0 |
| warmup_epochs | 10 |

**(b)** ViT-Base Configuration

| config | value |
|---|---|
| architecture | vit_base |
| n_prototypes | 65536 |
| patch size | 16 |
| momentum_teacher | 0.996 |
| drop_path_rate | 0.1 |
| bottleneck_dim | 256 |
| global_crops_scale | [0.32, 1.0] |
| local_crops_scale | [0.05, 0.32] |
| optimizer | AdamW |
| learning rate (lr) | $7.5e-4$ |
| weight_decay | 0.04 |
| weight_decay_end | 0.4 |
| freeze_prototypes | false |
| min_lr | $2e-6$ |
| clip_grad | 0.3 |
| warmup_epochs | 10 |

evidenced by the results in Section 4.4. We nonetheless view this as a meaningful contribution to the community, as it enables systematic evaluation of prototypical SSL methods across different degrees of collapse: whereas Govindarajan et al. (2024) reduces collapse, our approach eliminates it entirely[7], providing a complementary perspective for understanding its role in representation learning.

## B  TRAINING DETAILS

### B.1  HYPERPARAMETERS

For our experiments in Sections 4.3 and 4.4 our baselines DINO, DINO + KP and CARP are trained using a ViT-S/16 architecture while CARL is trained using a ResNet-50 backbone. Additionally, for CARL we increase the number of initialized prototypes to 65536 to enable a fairer comparison and adopt the same training improvements incorporated in CARP, which includes multi-crop augmentation and added schedulers. Besides accommodating for these architectural changes in CARL and CARP, the rest of the official codebase is left untouched. For the DINO model we use the officially released codebase without applying any changes (Caron et al., 2021). For DINO+KP, we use the same codebase as for DINO with a minor modification: we incorporate the KoLeo-Prototype regularization proposed by Govindarajan et al. (2024), setting the regularization strength to the empirically validated value of 0.1 reported in their paper.

We report the hyperparameters used to train the vision transformer backbones (Dosovitskiy et al., 2021) with CARP and CARP + Decoupling in Table B.1. In addition to these hyperparameters, CARP + Decoupling employs a weight threshold of 0.3 for the resurrect strategy, which is relatively high given the 65536 components in the mixture. Furthermore, for the smoothing factor $\beta$, we use a linear scheduler that increases from 0.1 at the start of training to 0.5 at its conclusion.

### B.1.1  EXPERIMENT: UNIQUENESS OF PROTOTYPES UNDER DIFFERENT THRESHOLDS

In our experiment described in Section 4.2, as well as in the reported number of unique prototypes in Table 1, we use the officially released pre-trained weights and reported numbers for all methods except DINOv2. At the time of writing, the prototype weights for DINOv2 were not publicly available; therefore, we rely on the re-produced weights of a ViT-L/14 model from Venkataramanan et al. (2025). For DINO and CARP, we evaluate their ResNet-50 checkpoints trained for 400 epochs. For CAPI, we evaluate its ViT-L/14 checkpoint pre-trained on ImageNet-1k. Since the teacher branch's prototypes

---

[7]Under all thresholds used to define unique prototypes in our evaluations.

are unavailable, we instead use the EMA-updated student branch prototypes. We evaluate the iBOT ViT-S/16 checkpoint, and the iBOT-vMF + KP unique prototype values are directly extracted from Govindarajan et al.'s (2024) paper.

### B.1.2 EXPERIMENT: TRAINING DYNAMICS

We train all baselines and CARP + Decoupling on ImageNet-1k (Deng et al., 2009) for 100 epochs using 6 local crops. For linear evaluation, we freeze the backbone and train a single linear classifier using a batch size of 1024 with PyTorch's LBFGS optimizer (Liu & Nocedal, 1989; Paszke et al., 2019). Concretely, we minimize cross-entropy on the training set using L-BFGS with `max_iter`=150, `tolerance_grad`=$1\times10^{-5}$, `tolerance_change`=$1\times10^{-9}$, and `history_size`=10.

### B.1.3 EXPERIMENT: EFFECT OF PROTOTYPE DIVERSITY ACROSS HEAD, MEDIUM AND TAIL CLASSES

We train all our baselines and CARP + Decoupling on iNaturalist2018 dataset using 10 local crops. All models use a ViT-S/16 backbone trained for 300 epochs on iNaturalist2018. The iNaturalist2018 challenge is a large scale species classification challenge that has long been the standard dataset for evaluating long-tailed performance. There are 8142 classes in this dataset, with 437,513 training images, and 24,426 validation images. We use the validation set for evaluation as the real test is deliberately left unavailable for public use. The dataset follows a heavy long-tail, with multiple classes appearing very few times in the training set. Following Liu et al. (2019), we consider all classes with more than 100 images each as *head* classes, between 20 to a 100 images each as *medium* classes, and less than 20 images each to be *tail* classes. Using this design choice, we have 842 head classes, 3701 medium classes, and 3599 tail classes.

We adopt the linear evaluation protocol of Caron et al. (2021) to assess all methods, with results reported in Table 2.

## C EXTENDED RESULTS

The preceding sections establish the empirical evidence motivating the proposed decoupled framework and its effects on prototype diversity, training dynamics, and performance across distribution regimes. These findings demonstrate that joint optimization induces a form of shortcut learning, driving prototypes toward redundant representations early in training, whereas decoupled updates maintain a higher prototype diversity.

### C.1 THEORETICAL PERSPECTIVE ON PROTOTYPE COLLAPSE

The empirical results consistently reveal a strong link between the way prototypes are optimized and the emergence of partial prototype collapse. These patterns point toward structural properties of the optimization that influence how prototypes distribute across the representation space. This section introduces a theoretical perspective that explains why joint updates promote redundancy and why separating prototype estimation leads to a diverse set of prototypes.

#### C.1.1 DECOUPLING AND PROTOTYPE COLLAPSE

Consider the prototype subproblem at fixed encoder parameters $\theta$ and soft assignments $\gamma_{ik}$ induced by the teacher-student logits. We can pose the subproblem as

$$J(C \mid \theta, \gamma) = \sum_{i,k} \gamma_{ik}\|h_i - c_k\|^2, \quad h_i = f_\theta(x_i),$$

whose unique minimizers are the responsibility-weighted centroids $c_k^* = \frac{\sum_i \gamma_{ik} h_i}{\sum_i \gamma_{ik}}$. If two distinct components $j \neq k$ have different responsibilities $\gamma_{:j} \neq \gamma_{:k}$, then $c_j^* \neq c_k^*$ almost surely; hence collapsed prototypes are not stationary points of the prototype subproblem $J$.

The decoupled optimization approximates a coordinate-descent / EM scheme. For fixed $\theta_t$ at step $t$ the GMM update performs an online M-step that pushes the full set of prototypes $C_t$ toward the centroids $c_k^*(\theta_t, \gamma_t)$. Moreover, for fixed $C_t$, we update $\theta$ on the SSL objective $\mathcal{L}_f(\theta, C_t)$. This keeps prototypes close to block-optimal solutions of the prototype subproblem at each step, which disfavors collapse whenever components have distinct assignments.

In contrast, joint gradient updates on $(\theta, C)$ only enforce joint stationarity of $\mathcal{L}_f(\theta, C_t)$, and do **not** enforce optimality of $J(C \mid \theta, \gamma)$. In other words, collapse with $c_j = c_k$ can be joint stationary points of the coupled system, even though they are suboptimal for the prototype subproblem. This is what we observe in our analysis of DINO-style modeling (Table 1, Figure 1a, Figure 4).

We note that **if** student prototypes of DINO exhibit collapse, the gradients for collapsed components receive identical updates, so collapsed prototypes cannot re-separate. Furthermore, any collapse in the student model will invoke a collapse in the teacher, whose prototypes are EMA updates of the student. Hence, teacher asymmetry does not meaningfully help prevent collapse.

### C.1.2 Formalisation of decoupling prototype collapse

**Lemma C.1** (Unique Centroids). For the prototype subproblem

$$J(C \mid \theta, \gamma) = \sum_k J_k(c_k), \quad J_k(c_k) = \sum_i \gamma_{ik} \|h_i - c_k\|^2, \tag{C.1}$$

each $J_k$ is strictly convex quadratic in $c_k$ with unique minimizer

$$c_k^*(\theta, \gamma) = \frac{1}{m_k} \sum_i \gamma_{ik} h_i, \quad m_k = \sum_i \gamma_{ik}. \tag{C.2}$$

*Proof.* The gradient and Hessian of $J_k$ are given by

$$\nabla_{c_k} J_k(c_k) = -2 \sum_i \gamma_{ik} h_i + 2 m_k c_k, \quad \nabla_{c_k}^2 J_k(c_k) = 2 m_k I_d. \tag{C.3}$$

As $m_k > 0$, the Hessian is positive definite, so $J_k$ is strictly convex with unique minimizer

$$\nabla_{c_k} J_k(c_k) = 0 \Rightarrow c_k^* = \frac{1}{m_k} \sum_i \gamma_{ik} h_i, \tag{C.4}$$

as we wanted to show. $\square$

**Proposition C.2** (Non-collapse for decoupled prototype subproblem). Assume features $h_i \in \mathbb{R}^d$ are drawn from a distribution that is absolutely continuous w.r.t. Lebesgue measure. Fix responsibilities $\gamma_{ik} \geq 0$ and masses $m_k = \sum_i \gamma_{ik} > 0$.

Then for any pair $j \neq k$ with $\gamma_{:j} \neq \gamma_{:k}$, we have that $\Pr(c_j^* = c_k^*) = 0$, i.e. $c_k^* \neq c_j^*$ almost surely. Equivalently, collapsed centroids occur on a subset of measure zero.

*Proof.* By Lemma 1, the centroids are

$$c_j^* = \frac{1}{m_j} \sum_i \gamma_{ij} h_i, \quad c_k^* = \frac{1}{m_k} \sum_i \gamma_{ik} h_i.$$

These are convex combinations of the features with normalized weight distributions $(\gamma_{ij}/m_j)_i$ and $(\gamma_{ik}/m_k)_i$.

Since $\gamma_{:j} \neq \gamma_{:k}$, these weight distributions differ: there exists at least one index $i^*$ such that $\gamma_{i^*j}/m_j \neq \gamma_{i^*k}/m_k$.

Collapse $c_j^* = c_k^*$ occurs if and only if

$$\sum_i \frac{\gamma_{ij}}{m_j} h_i = \sum_i \frac{\gamma_{ik}}{m_k} h_i \quad \Leftrightarrow \quad \sum_i w_i h_i = 0,$$

where $w_i = \gamma_{ij}/m_j - \gamma_{ik}/m_k$.

Since the weight distributions differ, $w \neq 0$ (i.e., not all $w_i = 0$). The collapse condition $\sum_i w_i h_i = 0$ therefore imposes a non-trivial linear constraint on the features.

Geometrically, this defines a hyperplane in $\mathbb{R}^{n \times d}$; the set $\{(h_1, \ldots, h_n) : \sum_i w_i h_i = 0\}$ is a $(nd - 1)$-dimensional affine subspace of the ambient $nd$-dimensional space. Since each $h_i$ is drawn independently from a distribution absolutely continuous with respect to Lebesgue measure, the joint distribution of $(h_1, \ldots, h_n)$ is absolutely continuous on $\mathbb{R}^{n \times d}$, and assigns measure zero to any proper affine subspace.

Therefore, $P(c_j^* = c_k^*) = 0$. □

**Corollary C.3** (No Prototype Collapse at Decoupled Fixed Points). Let $(\theta^*, C^*)$ be a decoupled fixed point, such that

- $C^*$ is a local minimizer of the prototype subproblem

$$C^* = \arg \min_C J(C \mid \theta^*, \gamma^*),$$

- $\theta^*$ is a stationary point of the SSL objective

$$\nabla_\theta \mathcal{L}_f(\theta^*, C^*) = 0.$$

Then given the conditions of Proposition 1, for any pair of distinct components $j \neq k$ with $\gamma_{:j}^* \neq \gamma_{:k}^*$, we have

$$P(c_j^* = c_k^*) = 0.$$

In other words, decoupled fixed points do not admit prototype collapse almost surely whenever the induced responsibilities are non-identical and non-degenerate.

*Proof.* Since $C^*$ minimizes $J(\cdot \mid \theta^*, \gamma^*)$, each centroid must be the responsibility-weighted mean:

$$c_k^* = \frac{1}{m_k^*} \sum_i \gamma_{ik}^* h_i^*.$$

By Proposition 1, if $\gamma_{:j}^* \neq \gamma_{:k}^*$ and $m_j^*, m_k^* > 0$, then

$$P(c_j^* = c_k^*) = 0.$$

□

Note that this result does not necessarily hold for coupled optimization such as with DINO, which can yield cases where a joint optima over the SSL loss function admits $c_j = c_k$.

## C.2 Linear Classification

### C.2.1 ImageNet-1k

In Table C.1, we report extended linear classification results obtained using our proposed decoupling approach built on top of CARP and compare them with other prototypical methods, including those with added dense objectives. For these linear results we train all our models using 10 local crops. For our k-NN evaluation we adopt Caron et al.'s (2021) evaluation protocol. We find that linear classification performance is highly sensitive to hyperparameter choices, consistent with observations by Caron et al. (2021). When directly applying DINO's (Caron et al., 2021) linear evaluation protocol, our results were substantially lower than suggested by our k-NN evaluations. We hypothesize that this protocol's hyperparameters are tuned for methods exhibiting strong prototypical collapse. This interpretation is supported by the fact that DINO and iBOT (Zhou et al., 2022)

**Table C.1:** ImageNet evaluation results for various prototypical SSL methods and architectures.

| Method | Backbone | Epochs | k-NN (%) | Linear (%) |
|---|---|---|---|---|
| DINO | RN-50 | 400 | 67.5 | 75.3 |
| SWAV | RN-50 | 400 | 65.0 | 74.6[†] |
| CARP | RN-50 | 400 | 67.7 | 75.3 |
| CARP + Decoupling | RN-50 | 400 | 69.1 | 75.3 |
| DINO | ViT-S/16 | 300 | 72.8 | 76.2 |
| iBOT | ViT-S/16 | 300 | 74.6 | 77.4 |
| CARP | ViT-S/16 | 300 | 73.6 | 76.3 |
| CARP + KP | ViT-S/16 | 300 | 73.7 | 76.1 |
| CARP + Decoupling | ViT-S/16 | 300 | 74.1 | 76.2 |
| MSN | ViT-S/16 | 600 | – | 76.9 |
| iBOT | ViT-S/16 | 800 | 75.2 | 77.9 |
| DINO | ViT-S/16 | 800 | 74.5 | 77.0 |
| DINO-vMF | ViT-S/16 | 800 | 74.7 | 77.0 |
| iBOT-vMF | ViT-S/16 | 800 | 75.3 | 77.9 |
| iBOT-vMF + KP | ViT-S/16 | 800 | 75.3 | 77.9 |
| CARP + Decoupling | ViT-S/16 | 800 | 75.3 | 76.4 |
| DINO | ViT-B/16 | 400 | 76.1 | 78.2 |
| DINO-vMF | ViT-B/16 | 400 | 77.4 | 78.8 |
| iBOT | ViT-B/16 | 400 | 77.1 | 79.5 |
| iBOT-vMF | ViT-B/16 | 400 | 78.7 | 80.3 |
| iBOT-vMF + KP | ViT-B/16 | 400 | 78.8 | 80.5 |
| CARP + Decoupling | ViT-B/16 | 400 | 76.7 | 78.1 |

**Table C.2:** iNat-2018 classification accuracies with full data. [†]Results retrieved from Govindarajan et al. (2024).

| Method | Unique Protos. | | Linear (%) | Fine-tuned (%) |
|---|---|---|---|---|
| **ViT-Small/16** | | | | |
| DINO-vMF[†] | 1380 | (2.1%) | 49.7 | 68.5 |
| iBOT-vMF[†] | 1804 | (22.0%) | 50.1 | 69.4 |
| iBOT-vMF (kd)[†] | 1843 | (22.5%) | 50.5 | 69.1 |
| iBOT-vMF (kp)[†] | 7895 | (96.4%) | 51.1 | 69.3 |
| MSN ($\lambda = 1$)[†] | 3363 | (41.3%) | 53.8 | 63.5 |
| PMSN ($\lambda = 5$)[†] | 3005 | (36.9%) | 41.8 | 64.2 |
| CARP + Decoupling | 65536 | (100%) | 49.1 | 71.7 |
| **ViT-Base/16** | | | | |
| iBOT-vMF (kd)[†] | 1634 | (19.9%) | 50.4 | 73.3 |
| iBOT-vMF (kp)[†] | 7573 | (92.4%) | 51.4 | 74.0 |
| CARP + Decoupling | 65536 | (100%) | 48.3 | 71.8 |

use these hyperparameters successfully, whereas methods with greater prototype diversity (Darcet et al., 2025; Oquab et al., 2024) typically perform much broader hyperparameter searches for linear evaluation — ranging from 30 configurations (Darcet et al., 2025) to over one hundred (Oquab et al., 2024). Accordingly, we adopt the lighter grid-search protocol of CAPI (Darcet et al., 2025) while retaining DINO's choice of which outputs to train the classifier on: the concatenation of the last four ViT layers for ViT-Small and the last ViT layer with averaged patch tokens for ViT-Base. While we believe further hyperparameter tuning could improve the linear results, we intentionally refrain from doing so and instead recommend interpreting our representations primarily through the k-NN metric, which offers a more reliable, fine-tuning-free measure of representation quality.

### C.2.2 iNATURALIST2018

In Table C.2, we report the linear and fine-tuned classification accuracies on iNaturalist2018 for CARP + Decoupling trained with 10 local crops over 300 epochs. Baseline results are taken from Govindarajan et al. (2024).

**Table C.3:** Sensitivity analysis of DINO under different configurations of hyperparameters.

| Method | Dec. | Cent. | SK | # Prototypes | Teacher temperature | k-NN (%) |
|--------|------|-------|-----|--------------|---------------------|----------|
| DINO | | ✓ | | 65536 | $0.04 \rightarrow 0.07$ | $68.9^{\dagger}$ |
| DINO | ✓ | ✓ | | 65536 | $0.04 \rightarrow 0.07$ | $27.8_{\downarrow\ 41.1}$ |
| DINO | ✓ | | ✓ | 65536 | $0.04 \rightarrow 0.07$ | $62.2_{\downarrow\ 6.7}$ |
| DINO | ✓ | | ✓ | 65536 | $0.05 \rightarrow 0.025$ | $67.7_{\downarrow\ 1.2}$ |
| DINO | ✓ | | ✓ | 1024 | $0.05 \rightarrow 0.025$ | $68.0_{\downarrow\ 0.9}$ |
| DINO | ✓ | | ✓ | 2048 | $0.05 \rightarrow 0.025$ | $68.7_{\downarrow\ 0.2}$ |

[†] Our re-produced DINO backbone using 6 local crops and mixed precision training.

## C.3 SENSITIVITY OF DINO WITH FULL DECOUPLING

While the proposed decoupling method mitigates prototypical collapse in both CARP (Silva & Ramírez Rivera, 2023) and DINO (Caron et al., 2021), it also introduces training instabilities in DINO. To examine DINOs sensitivity to increased prototype diversity, we reproduce a baseline DINO model using the official codebase with two slight modifications. First, we adapt the dataset loader to meet the constraints of our High-Performance Computing (HPC) environment, which prevents us from reusing the exact same data-loading seed. Second, to reduce training time and computational cost, we employ mixed-precision training instead of float32.

The resulting k-NN performance is reported in Table C.3. While the original DINO paper reports a k-NN accuracy of 69.7, our reproduced model exhibits a modest performance drop of 0.8 percentage points. We attribute this difference primarily to (a) the altered data ordering introduced by our changed dataloader and (b) the reduced numerical precision during training. All other baselines are produced under the same conditions, ensuring that the results in the table remain directly comparable. All models are trained on ImageNet-1k for 100 epochs using six local crops.

Building on these observations, we next investigate how DINO behaves when its prototype optimization is replaced by the decoupled online GMM update. As shown in Table C.3, replacing the prototype update mechanism from gradient descent to an online GMM, while keeping all other hyperparameters fixed, results in a performance reduction of 41.1 percentage points. This suggests that the centering operation in the standard DINO formulation is insufficient to promote sufficiently diverse *assignments* when prototype diversity increases. To address this limitation, we employ Sinkhorn-Knopp normalization (Cuturi, 2013), which substantially improves assignment uniformity and reduces the performance gap with vanilla DINO to 6.7 percentage points. This effect on improving uniformity accross assignments was first identified by Ruan et al. (2023) and later adopted by Oquab et al. (2024) in the DINOv2 framework.

Next, we follow the recommendations of Ruan et al. (2023) and adopt their teacher-temperature scheduling strategy, which progressively increases the sharpening of the teachers assignment distribution and thus provides a stronger learning signal. We observe that this modification substantially benefits DINO under increased prototype diversity: the additional sharpening yields a further improvement of 5.5 percentage points, narrowing the remaining gap to the vanilla DINO baseline even more effectively.

Finally, we investigate whether adjusting the number of prototypes can further stabilize training under the decoupled formulation. We find that substantially reducing the prototype count to 1k yields a small additional improvement, leaving a remaining gap of 0.9 percentage points to the baseline. Exploring this further, we identify an intermediate setting with 2k prototypes as the most effective configuration: this choice closes the gap almost entirely, reducing the difference to only 0.2 percentage points.

We use these selected configurations for all DINO + Decoupling experiments in Section 4.3 and Section 4.2. For the long-tailed setting in Section 4.4, we retain the same hyperparameters but increase the number of prototypes to 8192 to match the 8142 classes in iNaturalist-18. Applying the settings tuned on ImageNet-1k to iNaturalist-18 leads to a substantial performance drop, as reported in Table 2. Although further tuning tailored to the long-tailed distribution could likely mitigate this behavior, such investigation falls outside the scope of this work.

Ultimately, we find the pronounced sensitivity of DINO to prototype diversity to be an interesting empirical observation. Although CARP is architecturally similar to DINO, it differs in four key

components: random partitioning to stabilize training, unnormalized projection-head outputs, entropy regularization in place of centering the teacher logits, and a loss formulation that favors more one-hot-like predictions. Any one of these design choices, or some combination thereof, may underlie CARPs ability to remain stable and benefit from increased prototype diversity. Determining which components are responsible for this robustness, and how they interact, represents a compelling direction for future research.

## D   DECLARATION OF USE OF LARGE LANGUAGE MODELS

During the preparation of this work, the author(s) used Large Language Models (LLMs) for grammar and spelling checks, paraphrasing and rewording, code writing, writing assistance and reviewing, as well as for identifying and summarizing related work. After using these services, the author(s) reviewed and edited all generated content as needed and take(s) full responsibility for the publication's content.

