# OpenReview forum: "Why Prototypes Collapse: Diagnosing and Preventing Partial Collapse in Prototypical Self-Supervised Learning"
_ICLR.cc/2026/Conference — ICLR 2026 Poster_

### Official Review · Reviewer_k3fq · 2025-10-29

**Soundness:** 3
**Presentation:** 4
**Contribution:** 3
**Rating:** 6
**Confidence:** 4

**Summary:**

This paper studies partial prototype collapse in prototypical self-supervised learning (SSL), where many prototypes become almost the same. The authors show that this happens because encoders and prototypes are trained together under the same loss, which encourages shortcut learning. To fix this, they propose a fully decoupled training strategy: prototypes are learned as Gaussian mixture components using an online EM update, separated from encoder optimization. The method removes prototype collapse and improves prototype diversity, k-NN accuracy, and robustness to long-tailed datasets.

**Strengths:**

- The paper explains well why prototype collapse matters and gives strong evidence that this issue is general across many SSL methods, not only DINO.
- The proposed decoupling is easy to understand, no need new hyperparameters, and is supported by strong experiments.
- The authors test several popular baselines (DINO, CARP, iBOT, CAPI) and evaluate both balanced and long-tailed datasets. The analysis of prototype uniqueness and training dynamics is very complete.
- The paper is well structured and the figures help a lot to understand the argument.

**Weaknesses:**

The reviewer is happy with the paper overall. Here is a few area that could be improved:
- The paper mostly gives empirical evidence, but not a full theoretical explanation of why joint optimization causes collapse. IMO, a simple gradient or convergence analysis would make the argument stronger.
- The improvements in accuracy are clear but not very large (around +1–2%). The contribution is more diagnostic than algorithmic.
- The method is mainly tested on CARP. It would be good to see results on other recent large-scale SSL frameworks.

**Questions:**

This paper reminds me of non-parametric or memory-based approaches in contrastive learning—such as MoCo, which keeps a memory queue of past embeddings, or OBoW and SwAV, which use fixed or slowly updated codebooks as pseudo-prototypes.

Could the authors clarify whether their decoupled update is conceptually related to these methods?
And is the proposed online EM procedure similar to maintaining a dynamic memory bank of cluster centers?

---

> ### Author Response · Authors · 2025-11-27
>
> > The reviewer is happy with the paper overall.
>
> We thank the reviewer for the time and effort dedicated to evaluating our manuscript. We appreciate the insights and are pleased to learn that the overall assessment of the paper is positive.
>
> > The paper mostly gives empirical evidence, but not a full theoretical explanation of why joint optimization causes collapse. IMO, a simple gradient or convergence analysis would make the argument stronger.
>
> We thank the reviewer for emphasizing this important limitation. In the revised version, we have added a theoretical analysis that examines the optimization dynamics underlying partial prototype collapse, including a gradient-based view of how the joint objective incentivizes redundant prototype updates. This new material is included in Appendix C.1 and complements our empirical findings.
>
> > The improvements in accuracy are clear but not very large (around +1–2%). The contribution is more diagnostic than algorithmic.
>
>
> Our objective with this work is to provide insights and a deeper understanding of the commonly occuring partial prototype collapse on an important family of methods, such as DINO [[1](https://openaccess.thecvf.com/content/ICCV2021/html/Caron_Emerging_Properties_in_Self-Supervised_Vision_Transformers_ICCV_2021_paper.html)]  and CARP [[2](https://proceedings.neurips.cc/paper_files/paper/2023/hash/7caf9d251b546bc78078b35b4a6f3b7e-Abstract-Conference.html)].  Based on these insights we provide a simple but effective approach of decoupling that demonstrates a solution for this problem alleviating collapse across all thresholds. While our proposal shows modest results when it comes to accuracy, it builds a foundation for future research and further work to understand and alliviate these shortcomings.
>
>
> > The method is mainly tested on CARP. It would be good to see results on other recent large-scale SSL frameworks.
>
> We appreciate the reviewer raising this point. In the revised manuscript, we have incorporated DINO into all key experimental analyses, including prototype uniqueness, training dynamics, and long-tailed robustness (Sections 4.2–4.4). Decoupling similarly eliminates prototype collapse in DINO , however, unlike CARP, DINO exhibits sensitivity to the increased prototype diversity, leading to optimization instabilities. We report an extended study of these effects and the associated hyperparameter interactions in Appendix C.3.

---

> > ### Author Response · Authors · 2025-11-27
> >
> > >This paper reminds me of non-parametric or memory-based approaches in contrastive learning—such as MoCo, which keeps a memory queue of past embeddings, or OBoW and SwAV, which use fixed or slowly updated codebooks as pseudo-prototypes.
> > Could the authors clarify whether their decoupled update is conceptually related to these methods?
> >
> > We thank the reviewer for the question and we would be happy to clarify how our method relates to these methods. As we propose the decoupling for both CARP and DINO in our revised manuscript, we will use DINO as a point of reference as it is a more widely known framework.
> > * **SWaV** [[3](https://proceedings.neurips.cc/paper/2020/hash/70feb62b69f16e0238f741fab228fec2-Abstract.html)] is very similar to DINO, in its core DINO can be considered an extension of SWaV, where they introduce a self-distillation framework. As such, SWaV also has a set of learnable prototypes, although, a much smaller set of 3k, a quick evaluation of the official weights shows that the prototype collapse problem is also prevalent in SWaV with roughly 1.7k unique prototypes (60% of initialized ones). While, we haven't trained SWaV in our experiments, we would expect that decoupling its prototypes would yield a similar effect as the one observed in DINO.
> >
> > * **MoCo** [[4](https://openaccess.thecvf.com/content_CVPR_2020/html/He_Momentum_Contrast_for_Unsupervised_Visual_Representation_Learning_CVPR_2020_paper.html)] operate on *sample-to-sample* alignment. The queue stores past embeddings to maintain a large set of negative examples and stabilize contrastive learning. *No learnable prototypes or cluster-level structure is maintained*. In contrast, prototypical SSL (DINO, CARP) aligns samples to a set of learnable prototypes, i.e., *sample-to-prototype* distributions, which is precisely where partial prototype collapse occurs.
> >
> > * **OBoW** [[5](https://openaccess.thecvf.com/content/CVPR2021/html/Gidaris_OBoW_Online_Bag-of-Visual-Words_Generation_for_Self-Supervised_Learning_CVPR_2021_paper.html)] is conceptually related to DINO in that both predict distributions over learned visual concepts, but they differ in how these concepts are maintained. In DINO, the prototypes are learnable parameters and are *distilled from the student into the teacher via EMA updates*. In contrast, OBoW employs a linear prediction head *solely* in the student that can be interpreted as prototypes, but the teacher’s visual words are not jointly optimized; instead, they are *updated independently using an online queue*. Thus, OBoW performs a form of partial decoupling similar to what has been analyzed for CAPI [6](https://arxiv.org/abs/2502.08769), where only one branch’s prototypes participate in gradient-based optimization.
> >
> > > And is the proposed online EM procedure similar to maintaining a dynamic memory bank of cluster centers?
> >
> >
> > A dynamic memory bank can serve a similar decoupling role, but differs in its non-parametric nature and typical FIFO-style updates [4], which limit the temporal window of information it can retain (due to practical hardware constraints). In contrast, our online EM procedure updates parametric prototypes rather than replacing stored instances, enabling concepts to be learned and preserved over longer time horizons. Moreover, only prototypes receiving sufficient responsibility during the expectation step are updated, ensuring that updates are driven by meaningful assignment signals rather than by recently observed samples.
> >
> > [1] Caron, Mathilde, et al. "Emerging properties in self-supervised vision transformers." ICCV 2021.
> > [2] Silva, Thalles, and Ramírez Rivera, Adín. "Representation learning via consistent assignment of views over random partitions." NeurIPS 2023.
> > [3] Caron, Mathilde, et al. "Unsupervised Learning of Visual Features by Contrasting Cluster Assignments." NeurIPS 2020.
> > [4] He, Kaiming, et al. "Momentum Contrast for Unsupervised Visual Representation Learning." CVPR 2020.
> > [5] Gidaris, Spyros, et al. "Obow: Online bag-of-visual-words generation for self-supervised learning." CVPR 2021.
> > [6] Darcet, Timothée, et al. "Cluster and predict latent patches for improved masked image modeling." TMLR 2025.

---

### Official Review · Reviewer_qYSn · 2025-10-29

**Soundness:** 2
**Presentation:** 3
**Contribution:** 2
**Rating:** 4
**Confidence:** 3

**Summary:**

The article is on the prototype collapse problem for prototypical self-supervised learning algorithms. In such algorithms, representations are organized around certain prototype points, where the prototype points are also trained. A major issue is that during training, these prototypes can merge or collapse into a smaller set. The article offers an empirically evidenced hypothesis for the cause of this problem as the joint estimation of the encoder parameters and prototype points based on the same loss. Then it proposes a solution where different loss functions are defined for the encoder and prototype points, with the prototype loss based on Gaussian mixture models. The article presents various numerical experiments to support its analytical hypothesis and the proposed solution.

**Strengths:**

- The article offers a practically valuable observation about the potential cause of the prototype collapse problem in relation to the joint optimization of encoder parameters and prototype points.

- It also provides a loss separation approach (for encoder and prototype points) that is demonstrated to be successful in avoiding the collapse problem.

- A clear measure of collapse is also introduced in the article.

**Weaknesses:**

The main weakness of the article is that it is entirely based on an empirical approach, lacking a satisfactory analytical foundation for both the cause and the proposed solution of the prototype collapse problem. Although the article is useful in terms of practical implementations, a stronger theoretical grounding is needed to make its arguments more convincing and provide deeper insights for a potential ICLR submission.

**Questions:**

- Line 96: Provide the full form for EMA-updated (Exponential Moving Average).

- Line 99: Not a proper matrix multiplication (is the transpose of C necessary?).

- Definition 2.1: Is there a constraint $\|v_m\|=1$?

- Is this a logical necessity: “If partial prototype collapse truly stems from the joint optimization of prototypes and encoders, then separating their updates should prevent the collapse”?

- Is it possible to provide some analytical guarantee that the use of separate losses avoids prototype collapse?

---

> ### Author Response · Authors · 2025-11-26
>
> > The main weakness of the article is that it is [...] lacking a satisfactory analytical foundation for both the cause and the proposed solution of the prototype collapse problem.
>
>
> > Is it possible to provide some analytical guarantee that the use of separate losses avoids prototype collapse?
>
> We thank the reviewer and agree with the stated concern; a theoretical analysis of prototype collapse and decoupling would ultimately give more insight into the nature of the problem, and improve the quality of our manuscript.
>
> Our strategy follows the hypothesis in the original draft; joint optimization of encoder and prototypes can yield collapse as *joint stationary points*, which decoupled EM updates avoid almost surely via block-coordinate descent. Importantly, joint gradient updates is prone to component collapse in mixtures, which decoupled optimization with EM style updates avoids [[1](https://proceedings.mlr.press/v115/zhang20a.html)].
>
> This property of EM is well known in the literature [2,3], and is a key reason why it is typically preferred over gradient descent for non-convex multimodal loss landscapes, such as mixture models.
>
> In response to the reviewer’s request, we have added a complete theoretical analysis. These additions appear in Appendix C.1, and all relevant modifications addressing your comments are marked in *purple* in the revised manuscript.
>
> > Line 96: Provide the full form for EMA-updated (Exponential Moving Average).
>
> We appreciate the reviewer’s observation regarding the missing expansion of the abbreviation. The term has now been expanded as in the revised manuscript.
> > Line 99: Not a proper matrix multiplication (is the transpose of C necessary?).
>
> We thank the reviewer for pointing out this typo. The transpose of $C$ is indeed unnecessary, and we have corrected the notation accordingly in the revised manuscript.
>
> > Definition 2.1: Is there a constraint $|v_m|=1$?
>
> We appreciate the opportunity to clarify this point. Since each $v_m$ is a part of the set $C$, and all $c_k \in C$ satisfy $|c_k| = 1$, the unit-norm property naturally extends to $v_m$.
>
> > Is this a logical necessity: “If partial prototype collapse truly stems from the joint optimization of prototypes and encoders, then separating their updates should prevent the collapse”?
>
> We sincerly thank the reviewer for pointing out this logical inconsistency. Indeed, eliminating joint optimization does not guarantee the removal of partial prototype collapse. We have revised the sentence to avoid implying logical necessity. The updated version now reads:
>
> "*If partial prototype collapse is influenced by the joint optimization of prototypes and encoders, then separating their updates may reduce the likelihood or severity of the collapse.*"
>
> [1] Zhang, Poupart, Trimponias 2020 - Comparing EM with GD in Mixture Models of Two Components
> [2] Bishop 2006 - Pattern Recognition and Machine Learning
> [3] Murphy, 2012 - Machine Learning: A Probabilistic Perspective

---

### Official Review · Reviewer_7bAf · 2025-11-01

**Soundness:** 3
**Presentation:** 3
**Contribution:** 2
**Rating:** 4
**Confidence:** 5

**Summary:**

Prototypical self-supervised representation learning (SSL) methods such as DINO suffer from a partial prototype collapse issue, which heavily limits the diversity of the learned prototypes. This paper argues that this issue is caused by the joint optimization of the encoders and the prototypes. Inspired by partially decoupled training in CAPI [1], this paper proposes a fully decoupled training method. Through empirical experiments on the CARP [2] SSL method, the paper shows that this eliminates prototype collapse. By using diverse prototypes, they claim to learn improved representations, as reflected in improved downstream performance.

**Strengths:**

- Prototypical SSL methods such as DINO have grown to become widely used and better understanding the partial prototype collapse issue sheds light on the inner workings of these methods and provides insights on how to further improve them.
- The development of the fully decoupled training methodology by using an online Gaussian Mixture Model to update the prototypes is interesting. This is a valuable contribution as this involves specific choice of techniques such as responsibility based forgetting to avoid collapse and other techniques to further improve performance.
- Decoupled training of the encoders and the clustering model/prototypes offers an interesting pathway to simplify and improve prototypical SSL pre-training like DINO.

**Weaknesses:**

**Major:**
- The partial prototype collapse occurs in many prototypical learning methods. The decoupled training is presented as a more general contribution, but the paper only experiments with CARP. So, it is unclear if the proposed method is indeed general.
- In Table 3, there is a direct comparison between CARP and CARP+Decoupling only for the Resnet-50 model. Can the authors produce similar results for the other backbones (ViT-S, ViT-B)? Otherwise, it is hard to identify the impact of the proposed decoupled training method.
- One prior approach to prevent partial prototype collapse is the KoLeo-Proto (KP) regularization [3]. But the paper does not directly compare against this. For example, this can be clarified in Table 2 by adding the results for DINO+Decoupling and CARP+KP. That can show both the generality of the decoupling approach and comparison with KP regularization, at least in the long-tail context.
- Since the paper mainly experiments with the CARP method, it is also worth comparing CARP+Decoupling with CARP+KP in some of the ImageNet experiments in Table 3.
- The best possible linear probing accuracy achievable using the learned representations is a useful indicator of the representation quality. The hyperparameter choice is indeed important but running this is trivial and not expensive.  See section B.3 in DINOv2 [4]. The forward pass through the model is done once and several linear classifiers with different hyperparameters are trained simultaneously. For a publicly available implementation, I refer the authors to the iBOT repo (https://github.com/bytedance/ibot/).
- The paper shows improved prototype diversity but does not sufficiently demonstrate how this translates to better representations. An important aspect of SSL is the transferability of learned representations. Currently, no transfer learning experiments are included. [3] found that the transfer learning performance was similar or worse when pre-training on ImageNet with KP regularization. Is this also the case with Decoupling? In that sense, it is especially important to evaluate this (for say, CARP vs CARP+Decoupling and if possible CARP+KP).
- The motivation for choosing CARP in section 4.1 is not clear. It is stated that CARP exhibits a higher degree of collapse and hence, a bigger challenge. But in section 2.2 (line 153) it is stated that "CARP achieves substantially higher prototype diversity than DINO" and in section 4.3 (line 368) it is stated that "CARP is far less prone to prototype collapse than the baseline CARL". These statements are conflicting. Wouldn't it make sense to experiment with DINO instead if the idea is to choose the method that exhibits the highest amount of collapse/challenge (see Table 1)?

**Minor:**
- On one hand, this simplifies some complexities in training prototypical SSL methods by not requiring centering, sharpening. But this introduces other techniques and hyperparameter choices in the online GMM step. Since the responsibility based forgetting is important to prevent collapse, how sensitive is the training to the choice of the forgetting factor $\eta$? An ablation experiment on this choice can be useful. Also, what is the value of $\eta$ in the paper?

[1] Darcet et al. “Cluster and Predict Latent Patches for Improved Masked Image Modeling.” TMLR 2025.

[2] Silva et al. “Representation learning via consistent assignment of views over random partitions.” *NeurIPS 2023*.

[3] Govindarajan et al. “On partial prototype collapse in the dino family of self-supervised methods.” BMVC 2024.

[4] Oquab et al. “Dinov2: Learning robust visual features without supervision.” TMLR 2024.

**Questions:**

- How much time and memory overhead does the online GMM add, including all the proposed techniques in Table A.1?
- Is the online GMM setup (techniques and related hyperparameters) the same for both ImageNet and iNeturalist2018 pre-training? Is it necessary to adapt them to different pre-training datasets?
- In Table 2, what is the backbone used for all the models?
- The kNN performance reported in Figure 3b does not match with the results reported in Table 3. Can the authors clarify if the results in Figure 3b are after 100 epochs (corresponding to Fugure 4) and what backbone model is used here?
- CARP uses prototypes that are divided into P blocks of size N_B each. The paper specifies the total number of prototypes. But how many partitions are used? Just to clarify, is the online GMM applied to each block separately or is it applied to the complete set of prototypes?

---

> ### Author Response · Authors · 2025-11-26
>
> We thank the reviewer for their time and for the extensive and thoughtful suggestions provided. We appreciate the reviewer’s engagement with our work. We have addressed the majority of the comments raised, and we believe that the manuscript has improved significantly as a direct result of this feedback. All revisions made in response to the reviewer’s comments are highlighted in *RedOrange* throughout the updated manuscript for clarity.
> > The partial prototype collapse occurs in many prototypical learning methods. The decoupled training is presented as a more general contribution, but the paper only experiments with CARP. So, it is unclear if the proposed method is indeed general.
>
> >Wouldn't it make sense to experiment with DINO instead if the idea is to choose the method that exhibits the highest amount of collapse/challenge (see Table 1)?
>
> In our initial experiments, we observed a noticeable drop in performance when applying our online GMM decoupling approach to DINO. We have since conducted an extensive set of experiments and successfully incorporated the method into DINO. Retaining vanilla DINO with its centering operation caused a substantial performance drop. However, as we show in Table C.3, replacing the regularization with Sinkhorn-Knopp regularization closed the gap with our reproduced DINO baseline. We further improved performance by sharpening the temperature and reducing the number of prototypes, following Ruan et al. [1], who noted that DINO is sensitive to increased prototype diversity, requiring adjustments to its hyperparameters. All these details, along with a discussion, have been added to Appendix C.3.
>
> While we did not surpass the baseline DINO, we substantially closed the gap by incorporating the suggestions of Ruan et al. [1] while maintaining full prototype diversity. We have included the DINO + Dec. method with 65k prototypes in the unique prototypes experiments, under varying $\epsilon$ values, as well as the training dynamics, as shown in Figures 2a and 3. Although an extensive hyperparameter search for DINO is beyond the scope of this paper, these results demonstrate that our decoupling method effectively mitigates the prototype collapse problem, with only a minor reduction in representation quality (-0.2 in k-NN accuracy) for DINO.
>
> As future work, we believe that a more thorough exploration of the teacher-temperature schedule could further improve DINO under increased prototype diversity. Additionally, understanding which components of CARP enable stable learning without such tuning, would offer valuable guidance for designing prototype-based methods.
>
> **Table C.3 Sensitivity analysis of DINO under different configurations of hyperparameters:**
> | Method | Dec. | Cent. | SK | # Prototypes | Teacher temperature | k-NN (%) |
> |--------|------|-------|----|--------------|-------------------|----------|
> | DINO   |      | ✓     |    | 65536        | 0.04 → 0.07       | 68.9     |
> | DINO   | ✓    | ✓     |    | 65536        | 0.04 → 0.07       | 27.8     |
> | DINO   | ✓    |       | ✓  | 65536        | 0.04 → 0.07       | 62.2    |
> | DINO   | ✓    |       | ✓  | 65536        | 0.05 → 0.025      | 67.7    |
> | DINO   | ✓    |       | ✓  | 1024         | 0.05 → 0.025      | 68.0     |
> | DINO   | ✓    |       | ✓  | 2048         | 0.05 → 0.025      | 68.7     |
>
> > In Table 3, there is a direct comparison between CARP and CARP+Decoupling only for the Resnet-50 model. Can the authors produce similar results for the other backbones (ViT-S, ViT-B)? Otherwise, it is hard to identify the impact of the proposed decoupled training method.
>
> > Since the paper mainly experiments with the CARP method, it is also worth comparing CARP+Decoupling with CARP+KP in some of the ImageNet experiments in Table 3.
>
> For a more thorough comparison, we have extended Table 3 by adding results from two additional ViT-S backbones trained on ImageNet for 300 epochs, one using CARP and one using CARP + KP.
> **Additions to Table 3:**
> | Method | Backbone | Epochs | k-NN (%) | Linear (%) |
> |------------|--------------|------------|---------------|-----------------|
> | CARP       | ViT-S/16        | 300        | 73.6          | 76.3            |
> | CARP + KP       | ViT-S/16       | 300        | 73.7         | 76.1            |

---

> > ### Author Response · Authors · 2025-11-26
> >
> > > One prior approach to prevent partial prototype collapse is the KoLeo-Proto (KP) regularization [3]. But the paper does not directly compare against this. For example, this can be clarified in Table 2 by adding the results for DINO+Decoupling and CARP+KP. That can show both the generality of the decoupling approach and comparison with KP regularization, at least in the long-tail context.
> >
> >
> > We thank the reviewer for highlighting this point. We have now added both DINO + Decoupling and CARP + KP to Table 2, along with their corresponding results. Both variants underperform their respective baselines, and we have updated the discussion accordingly. In brief, we observe that DINO is sensitive to the increase in prototype diversity introduced by our decoupling strategy. For CARP + KP, we hypothesize that the KP loss weight requires retuning for CARP, and that the default hyperparameter setting hinders learning. A more detailed analysis has been added to Section 4.4.
> > **Updated Table 2:**
> > | Methods           | Head  | Medium  | Tail  | All   |
> > |-------------------|-------------|-----------------------|------------|-------|
> > | DINO              | 55.2        | 46.2                  | 41.9       | 45.3  |
> > | DINO + KP         | 59.8        | 50.4                  | 45.0       | 49.0  |
> > | DINO + Dec. | 44.9        | 37.3                  | 33.1       | 36.2  |
> > | CARP              | 56.0        | 46.9                  | 42.5       | 45.9  |
> > | CARP + KP         | 55.5        | 46.4                  | 41.8       | 45.3  |
> > | CARP + Dec.   | 59.1        | 49.3                  | 45.9       | 48.9  |
> >
> >
> > > The best possible linear probing accuracy achievable using the learned representations is a useful indicator of the representation quality. The hyperparameter choice is indeed important but running this is trivial and not expensive. See section B.3 in DINOv2 [4]. The forward pass through the model is done once and several linear classifiers with different hyperparameters are trained simultaneously. For a publicly available implementation, I refer the authors to the iBOT repo (https://github.com/bytedance/ibot/).
> >
> > We appreciate the suggestion and agree that linear probing is an informative evaluation of learned representations. However, pursuing “best possible” linear-probe results can introduce variability due to sensitivity to probe design, augmentations, and hyperparameter settings, and prior work has noted that these choices—as well as variability across runs—can meaningfully affect outcomes [2]. To provide a clear and stable assessment of the features themselves, we therefore place primary emphasis on k-NN accuracy. For completeness and direct comparison, we also include linear-probing results following the protocol used in CAPI [3]. While this setup does not involve as extensive a hyperparameter sweep as DINOv2 [4], it remains more comprehensive than the configuration originally used for DINO.
> >
> >
> > > The paper shows improved prototype diversity but does not sufficiently demonstrate how this translates to better representations. An important aspect of SSL is the transferability of learned representations. Currently, no transfer learning experiments are included. [3] found that the transfer learning performance was similar or worse when pre-training on ImageNet with KP regularization. Is this also the case with Decoupling? In that sense, it is especially important to evaluate this (for say, CARP vs CARP+Decoupling and if possible CARP+KP).
> >
> > We thank the reviewer for highlighting the findings of Govindarajan et al. [5] regarding prototype diversity and transferability. In response, we have added transfer learning experiments to the main paper, following the evaluation protocol of Ericsson et al. [6]. Consistent with the observations in [5], *our results show that increased prototype diversity does not necessarily translate into improved transfer performance*. For brevity, results for Food, Flower, and SUN397 are omitted from the table below, but the full set of results is included in the updated manuscript in Table 4.
> > **Table 4:**
> > | Method   | Arch  | Epochs | Aircr | C101 | Cars  | Pets  | VOC2007 | Avg   |
> > |----------|-------|--------|-------|------|-------|-------|---------|-------|
> > | DINO     | RN50  | 400    | 59.95 | 90.91 | 65.92 | 89.04 | 84.32  | 78.78 |
> > | CARP     | RN50  | 400    | 61.03 | 91.66 | 64.21 | 90.25 | 84.51  | 79.06 |
> > | CARP+Dec.| RN50  | 400    | 58.50 | 92.07 | 67.28 | 91.11 | 84.55  | 79.18 |
> > | CARP     | ViT-S | 300    | 59.16 | 93.61 | 64.48 | 93.38 | 85.18  | 79.46 |
> > | CARP+KP  | ViT-S | 300    | 60.48 | 93.54 | 63.55 | 93.40 | 85.34  | 79.67 |
> > | CARP+Dec.| ViT-S | 300    | 61.78 | 92.91 | 63.90 | 94.13 | 85.35  | 79.66 |

---

> > > ### Author Response · Authors · 2025-11-26
> > >
> > > >The motivation for choosing CARP in section 4.1 is not clear. It is stated that CARP exhibits a higher degree of collapse and hence, a bigger challenge.
> > >
> > > We thank the reviewer for the comment. The original phrasing in our motivation section appears to have led to a misinterpretation of our reasons for choosing CARP. In our initial version, we wrote:
> > > >>First, it is an instance-based approach (CARP), *which we have shown to exhibit* a higher degree of prototypical collapse...”
> > >
> > > Our intent in this sentence was to emphasize that *instance-based approaches*, as a class of frameworks, tend to exhibit a higher degree of prototype collapse—*not to attribute this specifically to CARP*. To avoid this ambiguity, we have revised the text to clarify this distinction.
> > >
> > >
> > > > On one hand, this simplifies some complexities in training prototypical SSL methods by not requiring centering, sharpening. But this introduces other techniques and hyperparameter choices in the online GMM step. Since the responsibility based forgetting is important to prevent collapse, how sensitive is the training to the choice of the forgetting factor $\eta$? An ablation experiment on this choice can be useful. Also, what is the value of $\eta$ in the paper?
> > >
> > > Regarding the forgetting factor, we use a scheduler that starts at $0.89$ and gradually approaches $0.99$ as $t$ increases. We have added an ablation varying both the initial value and the rate at which the schedule approaches its upper bound. The results show that performance is more sensitive to the initial forgetting factor: starting substantially lower or higher than $0.89$ weakens representation quality due to overly aggressive or overly conservative early prototype updates. In contrast, modifying the rate of convergence toward $0.99$ has only a minor effect on accuracy. Full details and results are provided in Appendix A.2.
> > >
> > > In addition, we would like to clarify that while the decoupling mechanism effectively mitigates prototypical collapse, thereby preventing all prototypes from converging to the same representation, *it does not by itself prevent trivial assignment solutions*. Without a regularization mechanism to encourage diverse assignments, the encoder can still produce outputs that concentrate on a single prototype, leading to a degenerate solution.
> > >
> > > > How much time and memory overhead does the online GMM add, including all the proposed techniques in Table A.1?
> > >
> > > The online GMM *introduces only minimal computational overhead*. As shown in Table 6, the total training time increases by approximately 0.2 hours when all proposed components are enabled, while providing noticeable memory advantages as batch size increases.  At small batch sizes, memory usage is similar to the joint-optimization baseline, but as the batch grows, the decoupled update avoids storing prototype-related computation graphs and leads to increasingly larger savings. In short, the method reduces memory consumption at scale with virtually no effect on training time. You can find the detailed results in Section 4.7 of the manuscript.
> > > **Table 6:**
> > >
> > > | Method     | Epochs | Batch Size | Total Crops | Time (h) |
> > > | ---------- | ------ | ---------- | ----------- | -------- |
> > > | CARP       | 100    | 1024       | 12          | 37.9     |
> > > | CARP + Dec. | 100    | 1024       | 12          | 38.1     |
> > >
> > > > Is the online GMM setup (techniques and related hyperparameters) the same for both ImageNet and iNeturalist2018 pre-training? Is it necessary to adapt them to different pre-training datasets?
> > >
> > > Yes. For CARP with decoupling, we use the same online GMM configuration and hyperparameters for both ImageNet-1k and iNaturalist2018, and we do not observe a need to adapt them across datasets. In contrast, applying the same settings to DINO with decoupling results in a noticeable performance drop on iNaturalist2018. This indicates that while the GMM setup is largely dataset-agnostic for CARP, DINO is more sensitive to the choice of hyperparameters and likely requires additional tuning when transferred to distributions such as iNaturalist2018.
> > >
> > > > In Table 2, what is the backbone used for all the models?
> > >
> > > All models in Table 2 use a ViT-S/16 backbone and are trained for 300 epochs on iNaturalist2018. We have updated Appendix B.1.3 to include these experimental details.
> > >
> > > > The kNN performance reported in Figure 3b does not match with the results reported in Table 3. Can the authors clarify if the results in Figure 3b are after 100 epochs (corresponding to Fugure 4) and what backbone model is used here?
> > >
> > > Your observation is correct. The k-NN results in Figure 3b correspond to the *same backbones used in Figure 4*, all of which were trained for 100 epochs. All methods *use a ViT-S/16 backbone, except CARL*, which follows its original implementation and uses a ResNet-50 encoder. These details are provided in Appendix B, including Section B.1 (Hyperparameters) and Section B.1.2 (Experiment: Training Dynamics).

---

> > > > ### Author Response · Authors · 2025-11-26
> > > >
> > > > > CARP uses prototypes that are divided into P blocks of size N_B each. The paper specifies the total number of prototypes. But how many partitions are used? Just to clarify, is the online GMM applied to each block separately or is it applied to the complete set of prototypes?
> > > >
> > > > We follow the hyperparameters from the original CARP formulation, using blocks of size 512. With 65,536 prototypes, this corresponds to 128 blocks. The online GMM operates on the full prototype set rather than being applied separately to each block.
> > > >
> > > > [1] Ruan, Yangjun, et al. "Weighted Ensemble Self-Supervised Learning." ICLR 2023.
> > > > [2] Caron, Mathilde, et al. "Emerging properties in self-supervised vision transformers." ICCV 2021.
> > > > [3] Darcet, Timothée, et al. "Cluster and predict latent patches for improved masked image modeling." TMLR 2025.
> > > > [4] Oquab, Maxime, et al. "Dinov2: Learning robust visual features without supervision." TMLR 2024.
> > > > [5] Govindarajan, Hariprasath, et al. “On Partial Prototype Collapse in the DINO Family of Self-Supervised Methods.” BMVC 2024.
> > > > [6] Ericsson, Linus, et al. "How Well Do Self-Supervised Models Transfer?" CVPR 2021.

---

### Official Review · Reviewer_Lrqz · 2025-11-01

**Soundness:** 3
**Presentation:** 2
**Contribution:** 3
**Rating:** 6
**Confidence:** 3

**Summary:**

This work explores the pervasiveness of a type of shortcut learning in self-supervised methods called partial protytype collapse. The authors show not all methods exhibit equal susceptibility to partial protytpe collapse, noting that methods that have decoupled training objectives for prototypes suffer much less from this shortcut. Motivated by this finding the authors proposes learning prototypes using an EM approach that's separate from the main self-supervisd training objective.

**Strengths:**

- findings in Section 2 that not all methods suffer from partial prototype collapse is novel and nicely motivates the decoupled training method proposed.
- the authors connect this collapse to recent findings regarding the muted effect of data scaling in SSL methods (lines 388). This is a fundamental open question that is carefully studied here through the lens of prototypes.
- examining training dynamics of the prototype collapse in Section 4.3 is an insightful approach that goes beyond simply examining the final representation. This section offers several nice insights such as the fact that the collapse emerges early on in training.
- authors include a nice comparison for KNN and linear probing for both iNaturalist and ImageNet, showing decoupling outperforms baselines.
- authors include hyperparameters and training details in the appendix

**Weaknesses:**

- define prototype in the introduction, and provide a clear accessible explanation of the partial prototype collapse you're aiming to solve.
- as described in section 2.1 the setting you're working with is focused on image domain, make sure this is clearly defined early on in the paper so readers have the right expectation of the setting you're exploring.
- similarly, technial terms such as KP regularization also require even a brief explanation to make this work accessible to broader audience. The authors assume too much prior knoweldge in my opinion.
- what limitations does using a Gaussian Mixture Model impose on the learned algorithm? A discussion and exploration of the limitations this imposes is missing.
- the authors discuss the merits of CARP w/ Decoupling in Section 4.4 with respect to head and tail classes, but offer a discussion or explanation of why CARP w/ decoupling underperforms DINO + KP for medium tailed classes as shown in Table 2.
- I'd be curious to see scaling trends for CARP w/ decoupling given the discussion of scale in Section 4.4. Have the authors experimented with this?

**Questions:**

see above

---

> ### Author Response · Authors · 2025-11-26
>
> We thank the reviewer for their thoughtful and constructive feedback, and we are glad that they found the insights in our work intresting. In response to the reviewer’s comments, we have made several revisions to the manuscript, which are highlighted in *Rhodamine* for clarity.
> >define prototype in the introduction, and provide a clear accessible explanation of the partial prototype collapse you're aiming to solve.
>
> A description of the role of prototypes in the SSL framework has been added to the introduction, and the first mention of partial prototype collapse now includes a cross-reference to its formal definition.
>
> >as described in section 2.1 the setting you're working with is focused on image domain, make sure this is clearly defined early on in the paper so readers have the right expectation of the setting you're exploring.
>
> An early clarification that our work focuses on the image domain has been added to the first paragraph of the introduction.
>
> > similarly, technial terms such as KP regularization also require even a brief explanation to make this work accessible to broader audience. The authors assume too much prior knoweldge in my opinion.
>
> A more detailed explanation of KP regularization has been added to the Preliminaries section.
>
> > what limitations does using a Gaussian Mixture Model impose on the learned algorithm? A discussion and exploration of the limitations this imposes is missing.
>
> We thank the reviewer for pointing out this omission. We have now added a summarized discussion of the limitations introduced by using the online Gaussian Mixture Model to the main paper, along with a more detailed analysis in Appendix A.3.
>
>
> > the authors discuss the merits of CARP w/ Decoupling in Section 4.4 with respect to head and tail classes, but offer a discussion or explanation of why CARP w/ decoupling underperforms DINO + KP for medium tailed classes as shown in Table 2.
>
> We thank the reviewer for the question. Following Reviewer 7bAf’s feedback, we expanded the experiments in Section 4.4 and added two new baselines: DINO + Decoupling and CARP + KP. These results prompted several revisions to the discussion of Table 2.
>
> Our primary focus in this experiment was to analyze where, across class-frequency groups, prototype diversity produces improvements, with particular attention to tail classes given their prevalence in real-world data. We also sought to evaluate whether the insight from Govindarajan et al. [1], that prototype diversity aids long-tailed recognition, generalizes to other methodological frameworks such as CARP.
>
> While our aim was not to contrast DINO and CARP directly, the difference between DINO + KP and CARP + Dec. on medium-tail classes raises a natural question. Although we do not have a definitive explanation, we can still offer a plausible hypothesis. Table 1 shows that CARP retains substantially more prototype diversity than DINO (10.5% vs. 1.5% unique prototypes). Given the 8,142 classes in iNaturalist, DINO is more likely to benefit from an increased number of effective prototypes, which KP provides.
>
> However, our extended results indicate that excessive prototype diversity can hinder DINO unless its hyperparameters are retuned; Appendix C.3 analyzes this sensitivity. We also find that applying KP to CARP without adjusting the corresponding loss weight adversely affects CARP’s learning dynamics. This issue is discussed in the revised Section 4.4.
>
> **Table 2: Performance comparison on the iNaturalist 2018 dataset.**
> | Methods           | Head  | Medium  | Tail  | All   |
> |-------------------|-------------|-----------------------|------------|-------|
> | DINO              | 55.2        | 46.2                  | 41.9       | 45.3  |
> | DINO + KP         | 59.8        | 50.4                  | 45.0       | 49.0  |
> | DINO + Dec. | 44.9        | 37.3                  | 33.1       | 36.2  |
> | CARP              | 56.0        | 46.9                  | 42.5       | 45.9  |
> | CARP + KP         | 55.5        | 46.4                  | 41.8       | 45.3  |
> | CARP + Dec.   | 59.1        | 49.3                  | 45.9       | 48.9  |

---

> > ### Author Response · Authors · 2025-11-26
> >
> > > I'd be curious to see scaling trends for CARP w/ decoupling given the discussion of scale in Section 4.4. Have the authors experimented with this?
> >
> > Our primary motivation for using iNaturalist2018 was to evaluate the backbones in a setting that is both challenging and representative of naturally long-tailed distributions. Compared with more uniformly distributed datasets, iNaturalist2018 provides a realistic testbed for examining how prototype diversity behaves when class frequencies vary substantially.
> >
> > Fan et al. [2] explore scaling trends using the multi-billion-sample MetaCLIP 2.5B dataset, which, despite curation, remains ”relatively long-tail” [3]. We agree that investigating prototype diversity at this scale would offer valuable insights, particularly given the discussion in Section 4.4. However, reproducing such large-scale experiments would require computational resources beyond what we could support, and therefore falls outside the scope of the present work.
> >
> > For this reason, we center our analysis on iNaturalist2018 as a demanding yet computationally feasible benchmark. We appreciate the reviewer’s suggestion and view large-scale investigations of decoupling as a promising direction for future research.
> >
> >
> > [1] Govindarajan, Hariprasath, et al. “On Partial Prototype Collapse in the DINO Family of Self-Supervised Methods.” BMVC 2024.
> > [2] Fan, David, et al. "Scaling Language-Free Visual Representation Learning." ICCV 2025.
> > [3] Xu, Hu, et al. "Demystifying CLIP Data." ICLR 2024

---

### Meta-Review · Area_Chair_bTGT · 2026-01-05

**Summary:**

**Summary of Contribution** \
The paper considers of the problem of partial prototype collapse occurring in cluster-based self-supervised learning and
proposes a solution decoupling the training of the encoder parameters and the prototypes. Estimation of the prototypes
is performed by fitting a Gaussian mixture model.

**Summary of Concerns** \
All concerns raised by reviewers were properly addressed by the authors. A summary is provided below:
1. Lack of theoretical grounding to motivate the proposed solution (raised by reviewers qYSn, k3fq) **Soundness**. The authors have added a section in the Appendix analysing the stationary points of the joint objective and relating with existing results on coordinate gradient descent for mixture models.
2. The experimental analysis doesn’t showcase the generality of the approach (raised by reviewers 7bAf, k3fq) **Quality**. The authors substantially extended the experimental evaluation by (i) integrating their solution into an additional baseline method (DINO), (ii) including experiments with Transformer-based backbones, (iii) adding further regularization baselines, and (iv) incorporating a transfer learning evaluation. These additions significantly improve the breadth, rigor, and completeness of the experimental methodology.

**Decision** \
The paper represents a nice addition to the literature of cluster-based self-supervised learning and specifically in dealing with the partial collapse problem. The authors have diligently addressed all concerns raised by the reviewers, consequently improving the overall quality and soundness of the paper.

**Reviewer Concerns:**

All concerns were adequately addressed by the authors.

**Reviewer Scores:**

Reviewer Lrqz primarily raised clarification questions and offered constructive suggestions aimed at improving the clarity and completeness of the presentation. The reviewer’s score would have remained unchanged (6).

Reviewer 7bAf conducted a thorough and detailed review, raising important and constructive points to strengthen the experimental methodology and improve the empirical evidence supporting the generality of the approach. The authors addressed all concerns satisfactorily. As a result, the reviewer would have increased their score (to 6 or slightly higher).

Reviewer qYSn raised concerns regarding the lack of theoretical motivation. The authors addressed this issue by adding a dedicated section in the appendix. Consequently, the reviewer would have increased their score to (6).

Reviewer k3fq would have maintained their positive score (6), as resolving the partial collapse problem leads to a modest improvement in performance.

---

### Decision · Program_Chairs · 2026-01-26

Accept (Poster)